# Raw material choices and technical practices as indices of cultural change: Characterizing obsidian consumption at 'Mycenaean' *Quartier Nu*, Malia (Crete)

**Tristan Carter** [1,2]*, **Vassilis Kilikoglou**[3]

**1** Department of Anthropology, McMaster University, Hamilton, Ontario, Canada, **2** School of Earth, Environment & Society, McMaster University, Hamilton, Ontario, Canada, **3** National Center for Scientific Research "Demokritos", Athens, Attica, Greece

* stringy@mcmaster.ca

**Data Availability Statement:** The data that underpins the results of this study are made fully available in the tables submitted with this paper and the Supporting Information files.

## Abstract

This paper takes a practice-based approach to the study of cultural identity, focusing on how raw material and technical choices involved in the production of quotidian tools served to both reproduce, and reflect a social group's very way of being. We then consider the (dis)continuity of obsidian blade-making traditions from Middle–Late Bronze Age Malia (north-central Crete), i.e., before and after a period of island-wide destructions, and appearance of foreign elements believed to reflect the arrival of a population from the Greek mainland (Mycenaeans). Methodologically this involves an integrated, 'thick description' obsidian characterisation study to detail long-term cultural traditions, including the use of Neutron Activation Analysis (NAA) to source the raw materials of 36 artifacts. The results show a significant degree of continuity in the community's lithic traditions, suggesting that many of the innovative features at Malia can be interpreted in terms of local factions appropriating new and foreign means of social distinction, rather than wholescale changes in community composition.

## Introduction

Obsidian characterisation studies have enjoyed a major resurgence over the past 15 years [1, 2], more than half a century after the first sourcing methods were developed [3, 4]. Initially these studies' primary aims were to shed light on socio-economic structures [5, 6], map trade networks and cultural interaction spheres [7, 8], and to document the long-term exploitation histories of specific obsidian sources [9, 10]. More recently, the interpretative remit for such work has expanded to include using obsidian sourcing as a proxy for reconstructing early hominin cognitive development, social complexity, and mobility [11, 12], and to chart the routes along which people, things, and ideas moved [13–17]. Characterisation studies have also recently been employed by those seeking to discuss social identity via the reconstruction of cultural traditions / technical practices [18–21]; it is this line of enquiry that represents our project's intellectual point of departure.

**Funding:** TC - Project Award from the Institute for the Study of Aegean Prehistory [INSTAP] http://www.aegeanprehistory.net/ The funding agency played no role in study design, data collection and analysis, decision to publish, or preparation of the manuscript.

**Competing interests:** The authors have declared that no competing interests exist.

In using obsidian sourcing as a means of discussing past identities we follow the argument (based on ethno-historic research) that raw material and technical choices are learned at a young age as part of an individual's socialisation, and the way they come to understand 'how things should be' in their social group [22–24]. Cultural identity is thus both expressed and reconstituted in how individuals perform these traditions [25]. Over the past two decades we have been employing a more holistic form of characterisation study (detailed below) as a means of mapping distinct traditions of obsidian consumption through space, and time within the context of Eastern Mediterranean prehistory [13, 26–28]. It is the elucidation of these 'communities of practice' [29] that represents a powerful means of utilising our characterisation data, as following debates from the sociology of technology it can be argued that such common knowledge between populations implies a significant level of on-the-ground interaction, likely maintained through inter-marriage, trading partnerships and other intense socio-economic relations [30].

While much of our work has examined these regional traditions synchronically, this study takes a diachronic, site-specific perspective as a means of examining a period of alleged socio-cultural and demographic discontinuity.

## Background to the project

This study of technical traditions and culture change at Malia over the Middle-Late Bronze Age [MBA–LBA] forms part of a longer-term consideration of obsidian consumption practices in the prehistoric southern Aegean (Crete, the southern Greek mainland, western Anatolia, and the smaller island chains between [Fig 1]). The exploitation of obsidian by southern Aegean communities is known to have spanned the Upper Palaeolithic to Late Bronze Age (15th-2nd millennia cal. BC), the raw materials procured primarily from local island-based sources, the most important–for toolmaking—being Sta Nychia (Adhamas) and Dhemenegaki on Melos in the Cyclades [31–33]. The Giali A source in the Dodecanese is the next most significant archaeologically (Fig 1). While this distinctive spherulitic obsidian is a poor tool-making resource, it was favoured by Cretan MBA–LBA elites (including those at Malia) for the manufacture of prestige goods such as vessels and sealstones [34]. While the recovery of non-Aegean obsidian in the region is rare [16], tiny amounts of central Anatolian products from Göllü Dağ and Nenezi Dağ (Fig 1) have been found on Crete, at Malia in particular [35, 36].

This characterisation study forms part of a larger project initiated in 1999, its aims being to detail traditions of Aegean obsidian consumption in Crete throughout the Bronze Age (c. 3000–1200 cal. BC) via the perspective of two north coast communities: Malia and Mochlos (Fig 2). Here we detail the characterisation of 36 artifacts from *Quartier Nu*, a LBA III complex at Malia (Fig 3), following that of 60 from neighbouring MBA *Quartier Mu* [36].

The analysis of the *Quartier Nu* assemblage had several research goals. Firstly, we wished to contribute to debates surrounding alleged culture change on Crete in the second half of the 2nd millennium cal. BC. The LBA I–II/III period witnessed a major reconfiguration in Cretan material culture, iconography, and language, which many believe to be the result of a population influx from the Greek mainland, i.e., the alleged invasion of 'Minoan Crete' by 'Mycenaean' mainlanders and/or the island's takeover by members of the major Cretan palatial centre of Knossos [37]. In short, our aim was to examine raw material and crafting choices at Malia over this period as an index of socio-cultural (dis)continuity.

A second project aim was to discuss long-term traditions of obsidian consumption at a specific community, rather than the more common region-level diachronic analyses [38–40], by combining our *Quartier Mu* and *Nu* analyses with those by Bellot-Gurlet et al. [35] of the Early Bronze Age [EBA] artifacts from Malia, a span of at least 1200 years from EBA II to LBA III (for cultural terminology and absolute dates see Table 1). While such diachronic site-specific

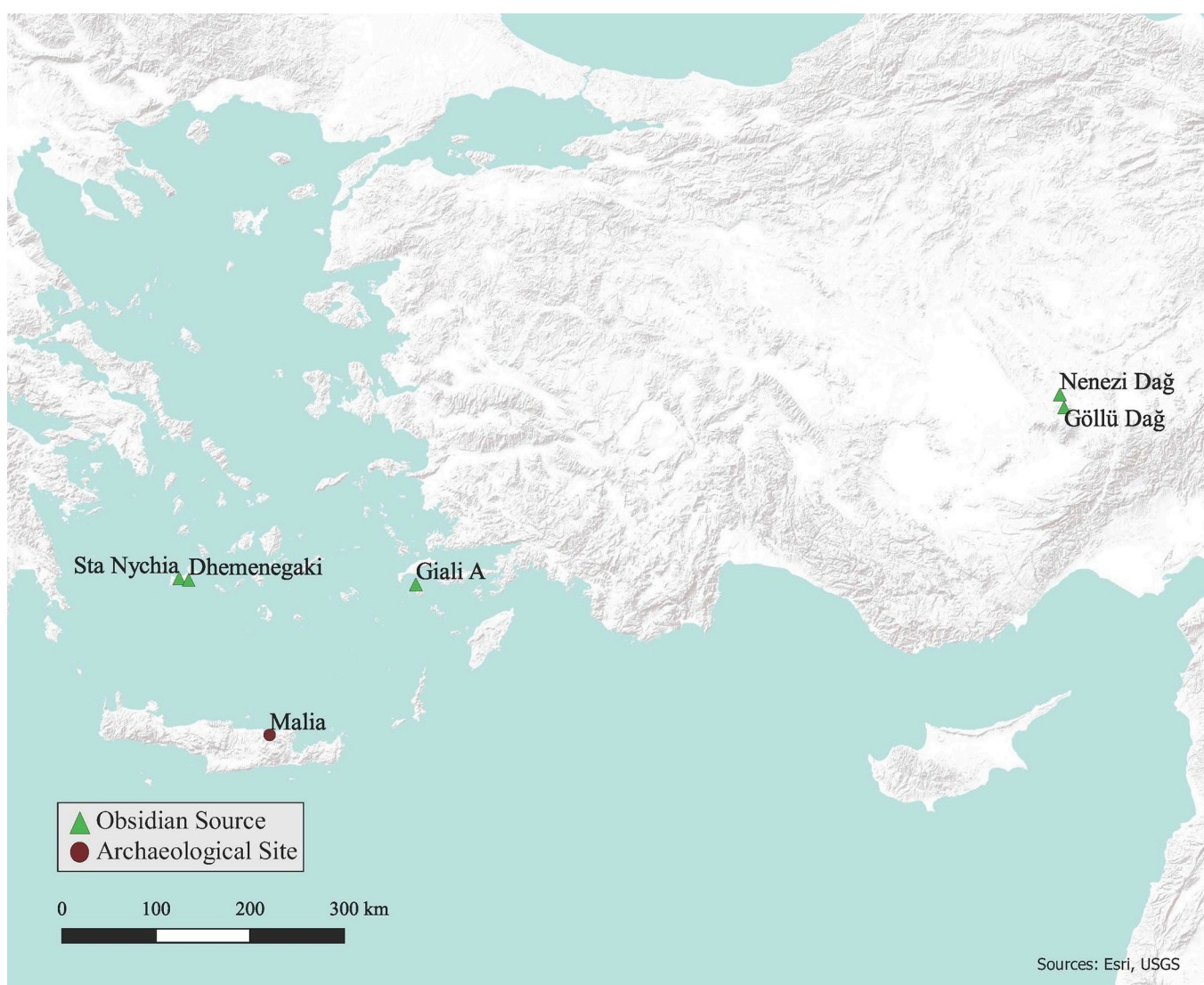

**Fig 1. Malia and the obsidian sources detailed in the study.** Compiled in QGIS 3.16.3 using ESRI World Terrain base map, by C. Lopez. Original copyright with the authors.

studies have been undertaken in other parts of the world [41, 42] this would be the first example from the Aegean.

Finally, with previous sourcing studies from Cretan sites suggesting that the Malia obsidian would be most likely Melian [27, 28, 36, 45–47], this study thus contributes to long-term history of these sources' exploitation, with the *Quartier Nu* material being the latest thus far analysed (second half of the 2[nd] millennium BC). A focus on the use history of individual sources is important, for while Melos has long been appreciated as the Aegean's primary supplier of obsidian [47], there has been limited attention paid to the relationship between the Dhemenegaki and Sta Nychia outcrops [48]; were they contemporaries, inter-dependant, or rivals?

## The nature of the problem: Cultural traditions of Late Minoan III Crete

Before turning to the site, assemblage, and analyses, a brief terminological and culture-historical background is required (for general overviews see [44, 49]). While the cultural name

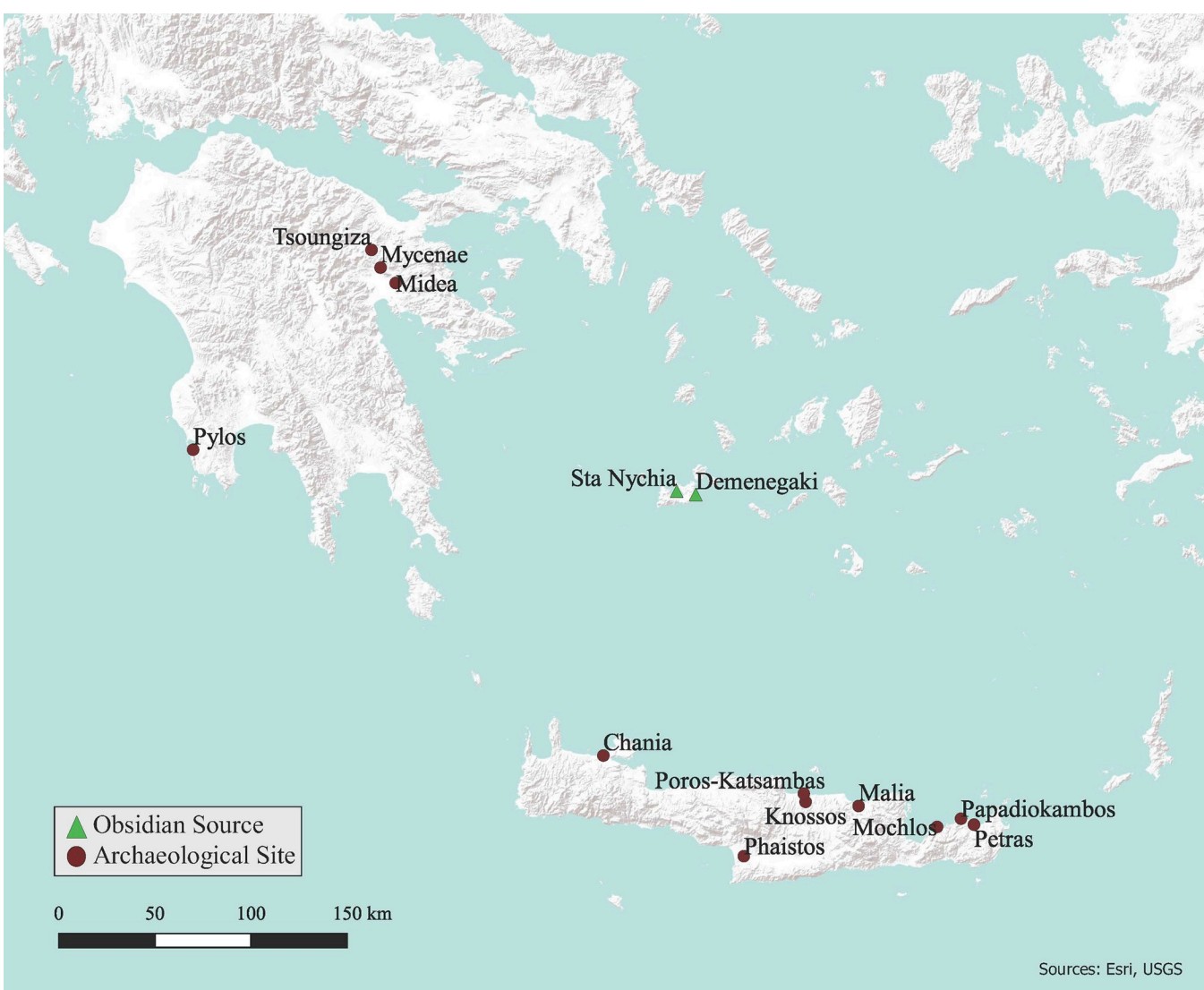

**Fig 2. Map showing main sites mentioned in the text.** Compiled in QGIS 3.16.3 using ESRI World Terrain base map, by C. Lopez. Original copyright with the authors.

'Minoan' has long been associated with the inhabitants of Bronze Age Crete (3rd-2nd millennia cal. BC), it is an invented term [50, 51], with these peoples' original name(s), language(s), and ethnic origin(s) remaining issues of debate [52, 53]. It is also employed as a chrono-cultural term, with the Early, Middle and Late Minoan periods representing the Early, Middle and Late Bronze Ages of Crete (EM, MM, LM = EBA, MBA and LBA respectively), as summarised in Table 1.

Another relative chronological scheme used to sub-divide Cretan Bronze Age archaeology centres upon the existence (or lack thereof) of 'palaces', i.e., the monumental structures that are traditionally viewed as the political centres of an early state system, the 'Minoan Civilization' [54]. The 'Prepalatial' period spans the 3rd millennium cal. BC (EM I-MM IA), after which the 'Protopalatial' (first-palace) period is defined by the construction of large, complex, court-centred buildings—or 'palaces'–at Knossos, Malia, Phaistos *inter alia* (Fig 2) at the start of 2nd millennium cal. BC (MM IB). Some 200 years later, major earthquake damage led to

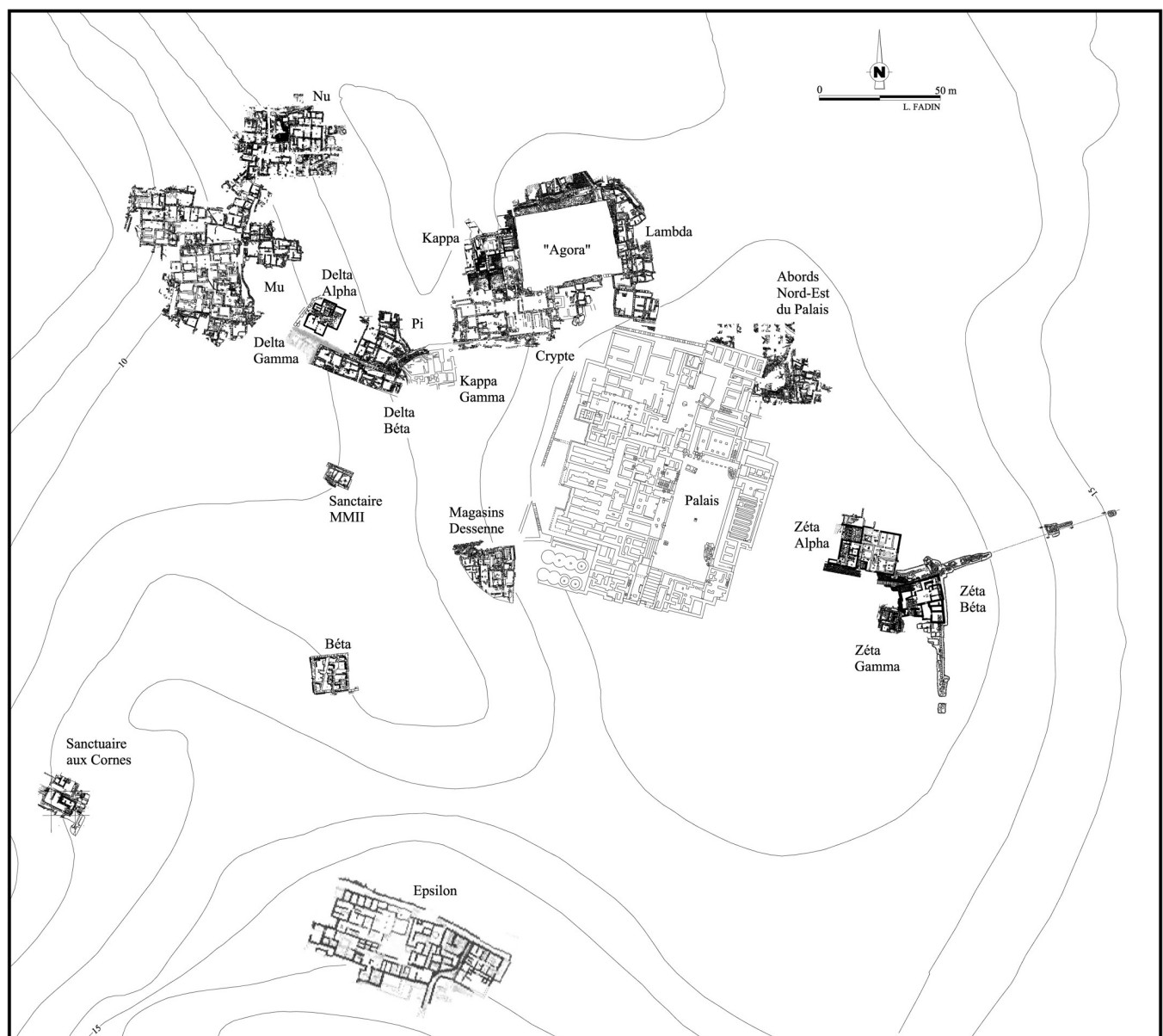

**Fig 3. Plan of the Malia excavations.** Plan reproduced with permission of the École Française d'Athènes.

major rebuilding work, and the construction of palaces anew throughout Crete [55], a phase referred to as the second palace, or 'Neopalatial' period (MM IIIA-LM IB).

It is what happens at the end of the Neopalatial period that sets the stage for the *Quartier Nu* study. In LM IA several sites went into economic, cultural, and demographic decline, the changes possibly triggered by a massive volcanic eruption on the island of Thera, 110 km to the north in the Cyclades [56]. Sometime later in LM IB (mid-2nd millennium cal. BC) we then witness the destruction and abandonment of nearly all other major palatial centres and middle ranking sites, this time apparently the result of aggressive human agents [57]. One site that seems to be largely unaffected by these destructions was Knossos, leading some to argue that Knossian expansionism was to blame for the island-wide destructions, with much of Crete

Table 1. **Minoan chronology, phasing and major events.** Based on [37, 43, 44].

| Periods | Ceramic Phases | Dates BC | History |
|---|---|---|---|
| Prepalatial | EM I–EM IIA | 3100–2200 | Regional centre at Knossos |
| | EM IIB–MM IA | 2200–1925 | Regional centres at Malia, Phaistos *inter alia* |
| Protopalatial | MM IB–MM IIA | 1925–1750 | First palaces / state level societies |
| | MM IIB | 1750–1700 | Earthquake destruction of First palaces |
| Neopalatial | MM IIIA–MM IIIB | 1700–1600 | Palace rebuilding, new palatial centres |
| | LM IA | 1600–1500 | Theran eruption |
| | LM IB | 1500–1450 | Decline/abandonment at many sites |
| Final Palatial | LM II | 1450–1400 | 1st phase of Mycenaeans at Knossos |
| | LM IIIA1 | 1400–1380 | 2nd phase of Mycenaeans at Knossos |
| | LM IIIA2 | 1380–1300 | Destruction of Mycenaean Knossos |
| Postpalatial | LM IIIB | 1300–1200 | Mycenean king at Chania? |

now conceivably under the rule of a single power, spanning the Final Palatial period of LM II-IIIA2 [58–61]. While one can talk of continuity regarding Knossos' palatial status, the nature of power seems to have undergone a significant transformation in LM II. This is evidenced by the appearance of a material culture, iconography, and burial practices that were hitherto associated primarily with populations of the mainland 'Mycenaean' culture. Knossos itself was eventually destroyed at the end of LM IIIA1, after which the character of its reoccupation took on an even greater Mycenaean nature, including the introduction of the Linear B script/language and the adoption of a new economic state-system in LM IIIA2-IIIB comparable to contemporary mainland palatial economies [62].

The basis of these socio-economic changes on Crete during the LM IB–LM IIIA2 period is a major topic of discussion. While some view the transition as the result of a Mycenaean invasion [63], others see the agents of change being local, with political instability enabling certain factions to develop power strategies that part-involved adopting value regimes (e.g., militarism), and elite symbols from the mainlander communities [64, 65]. Bioarchaeological approaches seem to support both positions. Thus, while a Y-chromosome DNA study (King et al. [66] suggested a latter 2nd millennium BC influx of population from the mainland a strontium ($^{87}$Sr/$^{86}$Sr) isotope and biodistance analysis of skeletons from stylistically Mycenaean 'warrior graves' at Knossos demonstrated clearly that the burials comprised a local population that had appropriated mainland funerary customs [67, 68]).

The LM II-IIIA1 period saw a post-destruction reoccupation of many sites across Crete, albeit never in the grandeur associated with their Neopalatial iterations [69]. These communities—including Malia and Mochlos–now had a strong Mycenaean character, particularly regarding their ceramic assemblages and burial practices. Such changes have led the excavators to ask the same identity questions posed by those working at Knossos, as neatly encapsulated in their articles' titles: 'Mycenaeans at Mochlos?' [70], and 'Mycenaeans at Malia?' [71]. A key aim of this paper is to contribute to these debates via a multi-faceted obsidian characterisation study to see if there is any significant difference between the crafting traditions of the Final / Postpalatial (LM IIIA2-IIIB) assemblages of 'Mycenaean(ized)' *Quartier Nu*, and those from the preceding 'Minoan' period of occupation, as represented by the datasets from *Quartier Mu* and *Batîment Pi* (Fig 3).

## Introduction to *Quartier Nu*

Located on the north coast of east-central Crete (Figs 1 and 2), the site of Malia is best known for its Middle and Late Bronze Age occupation when it was dominated by a monumental,

multi-functional elite building complex, the third largest of Crete's 'Minoan palaces' [43, 72]. Sometime around the end of the 'Late Minoan I B' [LM IB] period Malia's palace and associated settlement suffered a major destruction [43, 57]. This was followed by a period of reoccupation in the Final Palatial (LM II-III) period (Fig 3), as documented by excavations in the 'agora', plus *Quartiers Epsilon*, *Lambda*, and *Nu* [43], settlement clusters that may reflect different family clans [73].

Situated some 100 m west of the then defunct palace, *Quartier Nu* is situated atop a hillock with good access to the sea 450 m to the north-west (Fig 3) and comprises a large 750 m$^2$ complex organized around a small central court [74, 75]. While the court may have ideologically referred to the large central space of the abandoned palace, the architectural technology of *Quartier Nu* compares poorly with the fine buildings of Neopalatial Malia. Built in LM IIIA2, the complex was constructed on an area that was previously occupied from MM II–LM IA [76]. Parts of *Quartier Nu* suffered earthquake damage at the end of LM IIIA2, after which there was a second phase of occupation following some repairs and structural changes. The site was finally destroyed in early LM IIIB—the west wing by earthquake, the east wing by fire —its final phase of inhabitation perhaps spanning only two to three generations [73]. In total the LM IIIA2 –LM IIIB occupation would have spanned some 200 years across the 14th– 13th centuries BC [77]; alas no radiocarbon determinations are available. The site was then partly built on in the Venetian period (14th century AD), after which it seems to have lain fallow until the 1988–93 excavations.

Our clearest view of *Quartier Nu* is provided by its early LM IIIB plan, at which point the complex was arranged almost as a closed compound (c. 25 × 32 m) around an interior court (X11-X12), accessed from the north, with a small freestanding kitchen structure to the east (room XIV [Fig 4]). The court seems to have been the focus of ritual activity, as attested by a large house model and storage containers that sat atop a rare pebble mosaic, while pits to the south contained ceremonial deposits [73].

One question posed of *Quartier Nu* concerns the social units that dwelled there, and by extent the organisation of society more generally in Final Palatial Crete. Did the complex work as a single unit, or as a cluster of individual households? Artifact distribution suggests that the complex comprised an east and west wing that functionally duplicated each other, with bulk storage, serving and food/drink consumption attested in both [73, 78]. Small differences can be noted between the two, with grinding stones (food preparation) concentrated in the east, while the finest drinking (elite/male) vessels were restricted to the western half. *Quartier Nu* was thus likely inhabited by a single extended household, with a certain complementary relationship between the complex's two halves, one perhaps structured by gender roles and status, while the freestanding kitchen and courtyard were used for communal ceremonies [73]. It also produced evidence for industrial activities such as metalworking, dyeing and textile production, while the presence of four ceramic transport vessels inscribed with short Linear B texts suggest the hosting of visiting dignitaries, the wine containers acting as "monumentalized namecards", with the host-guest-gift relationship part underpinning a central and west Cretan elite network [79].

## Charting cultural traditions through characterisation studies

From the outset it should be appreciated that this project's aim was not to tackle issues of ethnicity *per se*, i.e. whether *Quartier Nu* was inhabited by 'Minoans' or 'Mycenaeans', as—following Barth [80]—we believe ethnicity to be a form of identity that is self-ascribed, and subjective (emic), rather than an objective category (etic) that can be read in the archaeological record on the basis of perceived cultural similarities and differences [81, 82]. Instead, we attempt to

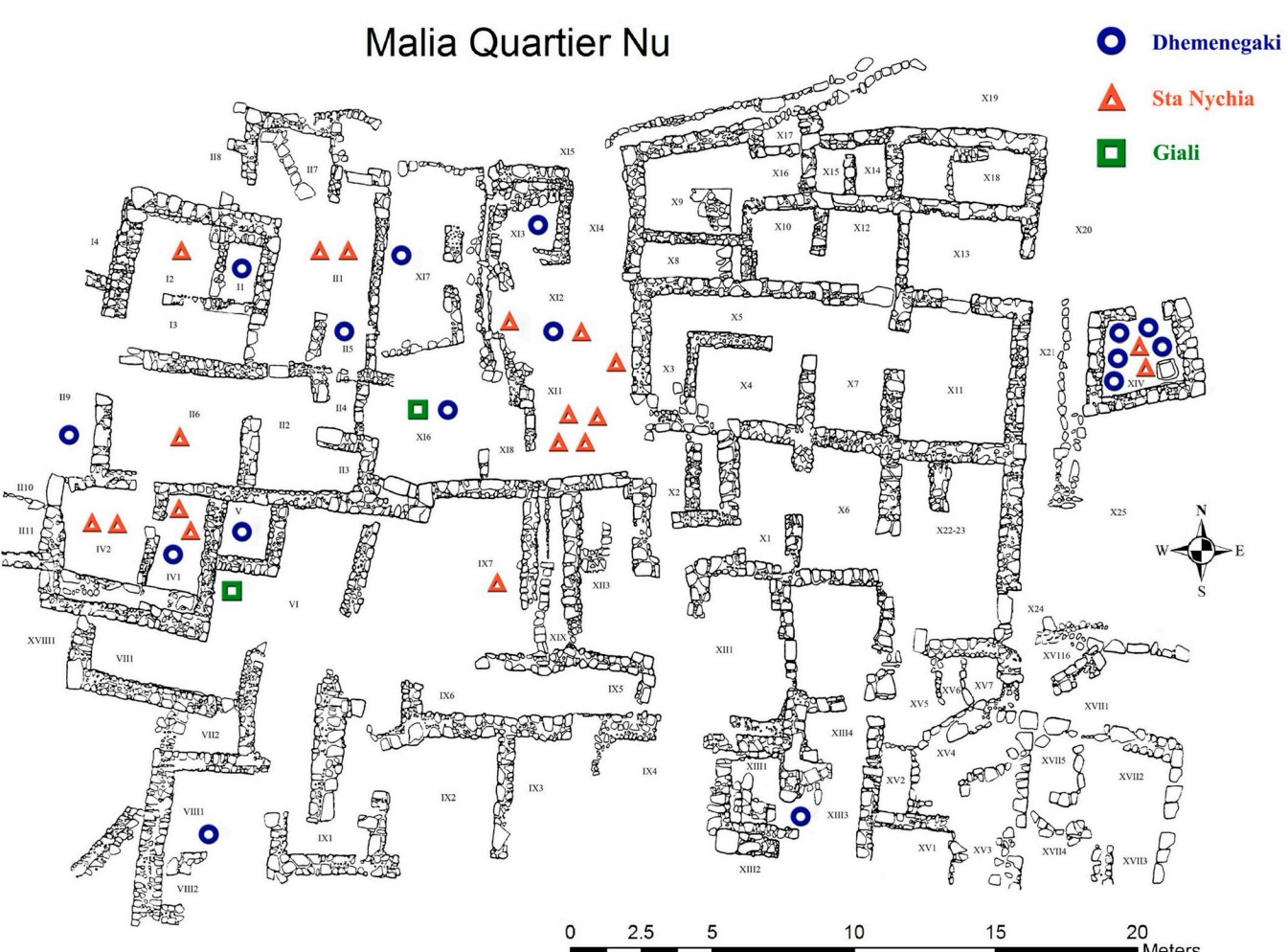

**Fig 4. Plan of *Quartier Nu* indicating distribution and source of artifacts characterized.** Plan reproduced with permission of the École Française d'Athènes, with additions by Kress, N, Milić, M.

reframe the debate in terms of detailing and contrasting cultural traditions over time. To achieve these aims we first consider this study's characterisation data diachronically at Malia, to see if raw material choices and technical practices remained constant, or changed over time, with a particular focus on the relationship between the Neopalatial (so-called 'Minoan') and Final / Postpalatial (allegedly 'Mycenaean') periods. We then discuss the *Quartier Nu* findings in relationship to broadly contemporary data from elsewhere, specifically Mochlos and Chania on Crete, plus Tsoungiza, and Midea on the Mycenaean mainland (Fig 2).

Methodologically our work necessarily involves moving beyond a consideration of raw materials alone; in short, we are interested in *characterisation*, not just sourcing. Historically Eastern Mediterranean characterisation studies have focused almost exclusively on an artifact's *composition*, with almost no other information provided on the object, aside from an occasional reference to 'blade', or 'flake', while illustrations and/or detailed typo-technological commentaries are rare if not absent (a notable exception being [83]). We have long argued that this approach is reductionist, leading us to develop an alternative method that reintroduces an archaeological sensibility [84]. 'Samples' are reconceptualised as 'artifacts' and are accorded a richer 'character' by considering not only their raw material (elemental

composition/source, visual and haptic qualities), but also how they were made, what they looked like, their spatial-temporal contexts and their prevalence in any given assemblage. Ultimately this serves to locate our sourcing studies within a *chaîne opératoire* analytical framework [85, 86], where one considers the various cultural choices involved in an artifact's life, from raw material procurement, via its technical-stylistic transformation, to use, and discard [87, 88]. Such an approach produces a more detailed and nuanced appreciation of obsidian consumption, with such 'thick description' [89] characterisation studies enabling us to elicit significant distinctions in how raw materials were circulated and worked through space and time [13, 14, 26, 27, 36, 84, 90, 91]. In short, this study focuses on practice, i.e., how people went about the manufacture of objects (the sequence of culturally informed raw material and technical choices), rather than just focusing on the raw material alone, or what the final product looked like [92, 93].

In many respects, the study of material culture as indices of identity has a long heritage in Aegean prehistory, albeit undertaken primarily through the lens of culture-history. This is most clearly seen in ceramic studies, implicit in terms such as 'Minoan pottery' [94], while certain forms of iconography, figurines, architecture, weaponry, and burial types have also been viewed as culture-specific signifiers [95, 96]. In this paradigm, the movement of such identity markers from a notional cultural/ethnic homeland into another region has traditionally been interpreted as evidence for population movement, the debate then revolving around its nature, reason, and scale, with migration, invasion, and colonisation all familiar explanatory tropes [63, 97–100].

In line with broader developments in anthropological archaeology [101–104], this familiar vision of 'pots = people' has been critically reappraised by Aegean prehistorians over the past 25 years, with arguments forwarded as to how foreign fashions and practices can be adopted by a culture, without an associated population movement [64, 65, 105]. Pertinent to this study for example, is how scholars have interpreted the appearance of the kylix (stemmed goblet) in Crete after the LM IB destructions, a classic 'Mycenaean' pottery form that has no antecedent in 'Minoan' assemblages. For some, this vessel type (along with other forms) is a key cultural marker for those who argue that these destructions and subsequent socio-economic changes were the result on an influx of mainlanders and/or their political influence [106]. If, however, one shifts from focusing on these vessels' form and decoration, to considering how they were made, then we note significant differences between mainland and Cretan kylikes, for while the stem of the former is solid, those made on Crete tend to be hollow [107]. Thus, rather than indices of a Mycenaean presence, the Cretan kylikes might be better viewed as a local emulation of an elite foreign drinking vessel (and attendant ceremonies), the goblets produced by Cretan potters who could replicate the form but were not cognisant of the original, mainland technological traditions of production. Such a claim does not necessarily negate the argument for an influx of Mycenaeans, but it does give pause for thought, and remind us of the potential complexity of social processes at play that go deeper than physical appearance alone.

In the above example we can see how a study of *practice*, rather than one based on form and decoration could provide significantly different interpretations as to an artifact's cultural association [105, 108]. In the Aegean such an approach has already been productively applied to several instances where the appearance of foreign material culture, burial practices and/or iconography had been interpreted as the result of population movement. Such studies are typically founded upon both a rich evidential bases, the data produced by detailed technical, typological, and contextual analyses, and a theoretical awareness concerning the relationships between practice and identity. While in certain instances such work has led to hypotheses that do not forefront migration to explain culture change [109], it is important to appreciate that these studies have *not* led to an across-the-board rejection of population movement as a

catalyst in socio-economic, technical, and/or artistic innovations [70, 110]. That said, one does see a shift towards discussions of much smaller scale forms of mobility, and 'communities of practice' to explain the emergence of inter-regionally shared cultural traditions [108, 111, 112].

Operationalising these theoretical premises involved a full technological, and typological analysis of the *Quartier Nu* assemblage prior to the selection of artifacts for sourcing; indeed, this study formed the basis of the sampling strategy. These typo-technological data were then integrated fully with source information generated by the NAA analysis, thus allowing us to detail specific modes of consumption by raw material. The study is also necessarily comparative, aiming to discriminate lithic traditions of Cretan (pre-LM IB destructions), and mainland populations, i.e., the notionally 'Minoan' and 'Mycenaean' ways of doing things. Given that pressure flaked blade production was the primary means of consuming obsidian throughout the Bronze Age Aegean [113], our methodology necessitates elucidating site-specific or regional distinctions in how this technique was articulated, as likely attested through the means in which the knappers prepared the core, initiated blade removal and rejuvenated flaking surfaces [114].

### The *Quartier Nu* obsidian assemblage

The *Quartier Nu* excavation generated 1306 chipped stone artifacts, 99% of which were made of obsidian (n = 1284). On their own, we cannot assign a date to this material beyond 'Bronze Age', with their chronological specifics dependent upon the associated ceramics (the full techno-typological study of this assemblage is being prepared for publication in the excavation monograph). Most of the 1284 pieces of obsidian came from deposits containing Final—Post-palatial pottery (n = 1146, 89%), though as to how much of this chipped stone is residual from earlier Neopalatial, or Protopalatial strata is impossible to say. The assemblage was also quite fragmented, with only five (small) blades recovered complete, the degree of breakage likely due to later disturbance and the fact that the site lays so close to the modern surface.

Most of the obsidian is grey black, relatively opaque and has a matte surface, visual characteristics long associated with the source products of Sta Nychia and Dhemenegaki on Melos [47], 160 km linear distance north-west of Malia (Figs 1 and 2). There were also a handful of lustrous black pieces of obsidian with white spherulites, typical of the Giali A source in the Dodecanese, 210 km north-east of Malia [34]. Finally, there were a few artifacts–mainly blades–whose translucency, colour, banding, and texture led them to be tentatively assigned a non-Aegean, likely central Anatolian source [16].

The *Quartier Nu* material thus appeared typical of Cretan Bronze Age assemblages, i.e., dominated by Melian obsidian used to make pressure flaked blades from unipolar cores, with most of the razor-sharp end-products being employed without further modification [113, 115] 2013, *inter alia*). Blades were typically flaked from half to two-thirds of the core's circumference, the back of the nucleus likely being covered by the knapper's hand and/or a simple stabilising device (Fig 5). Blade removal was initiated by either using a nodule's natural linear margin or by creating an artificial ridge of least resistance by cresting (Fig 5, 4–5). In the latter instance two crests were usually prepared, one either side of the face to be flaked; sometimes a posterior crest was flaked with rejuvenation in mind. The back of the core had often received some preparatory work, with flakes removed across to flatten it, the front of the nucleus ultimately gaining a more convex profile. If the core was made from a small rectangular nodule, then often the back would be left with a natural surface, the cortex then only being removed when it was necessary to revolve the nucleus 180 degrees to open a new flaking surface (Fig 5, 9). As such, some of the *Quartier Nu* cortical debris might be better viewed as rejuvenation flakes, rather than material pertaining to the initial stages of nodule shaping and core

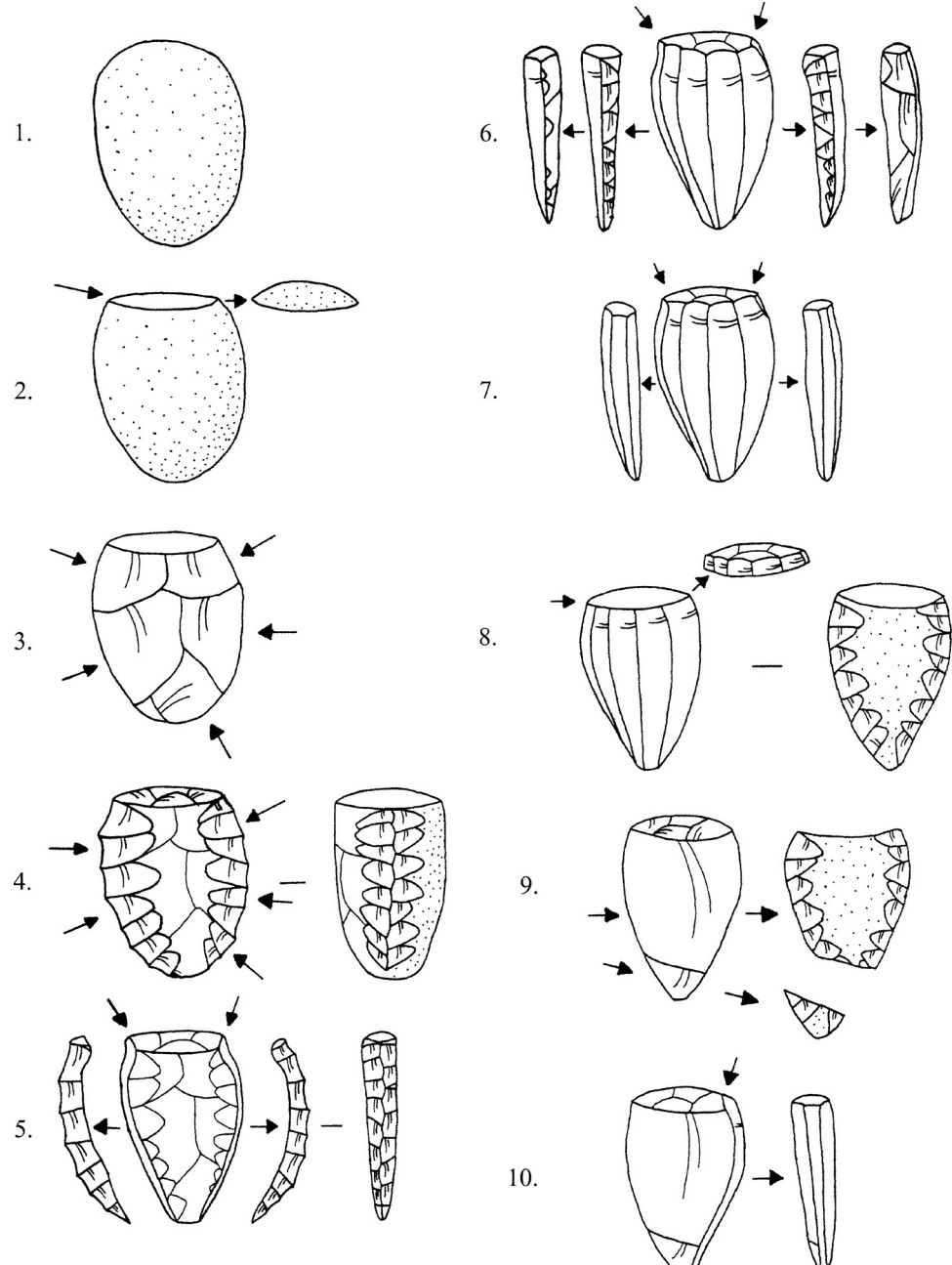

**Fig 5. General reconstruction of obsidian blade-core preparation and reduction sequence at Bronze Age Malia.**
Reproduced from [116]; original copyright with the authors.

preparation. The blades' distal profiles indicate that the cores had square bases, their original length approximately 3.5–4.5cm long based on the handful of complete end-products.

That part of the Final / Postpalatial obsidian assemblage visually identified as Melian (n = 1136/1146, 99%) comprises most of the manufacturing sequence–thus evidencing on-site blade production—with cortical waste, preparation and rejuvenation flakes, cores plus hundreds of broken blades (Fig 6). Some 11% of this assemblage had been modified (n = 121/1136), of which half were blades (n = 64), the remainder being blade-like flakes, and flakes

(n = 57); this retouched component included notched (occasionally denticulated) pieces, rudimentary perforators, plus a few end-scrapers, backed blades and trapezes. Tools were made and used throughout *Quartier Nu*, albeit at a relatively low-level, suggesting that we are dealing with a large degree of self-sufficiency in terms of obsidian access and technical competence. In most instances the material included one or two cores (often exhausted), broken end-products, and handfuls of secondary series (part-crested/cortical) blades, preparation, and rejuvenation flakes. In the western half of the complex such groups of material can be noted in the west, south and south-west wings, and within the main part of the building (rooms I1-I3, II1-II6, IV1-2, and VII1 [Fig 4]). In the structure's eastern half, one has much the same impression, with low levels of production attested in the south and south-east wing and the main rooms (X1-X12, X14, X22-X23). Production debris was also found in the external spaces to the north of *Quartier Nu*, while one of the greatest concentrations of material–including five blade cores–came from the central court. Finally, blades were also made in the stand-alone kitchen (room XIV).

Obsidian working at *Quartier Nu* thus seems to mirror what we see in the complex concerning food storage and consumption [73], i.e., relatively even distribution between the two halves of the structure, with no signs of inter-dependency. The integration of our information with other data sets will eventually provide a more refined vision of how these implements were being employed in domestic, craft and ritual activities.

## Sampling

Once a full typo-technological and contextual analysis had been undertaken of the *Quartier Nu* obsidian assemblage we chose 36 artifacts for elemental characterisation, just over 3% of the Final / Postpalatial dataset (n = 1146). Today, through access to more rapid, non-destructive XRF techniques [16, 28], we would select a considerably larger proportion of the assemblage for analysis, however, at the time NAA was the exclusive means of elementally characterising obsidian in Greece, whereby we were constrained sample-wise by the analytical costs.

The items were selected to represent raw material and typo-technological variability, the former with reference to colour, translucency, banding, inclusions, texture and cortex, the latter regarding the various stages of the pressure blade manufacturing sequence. The artifacts came from various contexts allowing us to consider intra-site/community distinctions in consumption (Table 2, Fig 4). Most of this material came from deposits associated with LM IIIB pottery (Postpalatial), aside from one blade that was accidentally selected from an LM IA—Neopalatial–deposit (MAN26).

Each artifact was given its own letter-numeric code: MAN01 –MAN36 (MAN = Malia Nu), and prior to the elemental analysis a provenance was suggested for the object's raw material on the basis visual characterisation (Table 2), to see if it would be possible in the future to visually source obsidian at Malia, and elsewhere without using what is a relatively expensive and part-destructive analytical technique [117, 118].

## Characterisation by NAA: Protocols, and results

Neutron Activation Analysis [NAA] was first used in Aegean obsidian characterization studies in the early 1970s, enabling the discrimination of not only the region's raw materials (from Melos, Giali and Antiparos), but also those from the Carpathians to the north, and central Anatolia to the east. Source discrimination was achieved through reference to their relative concentrations of Cs, Ta, Rb, Th, Tb, Ce and Fe, as expressed relative to the concentration of Sc [119]. NAA was also the first technique that was able to elementally distinguish the two

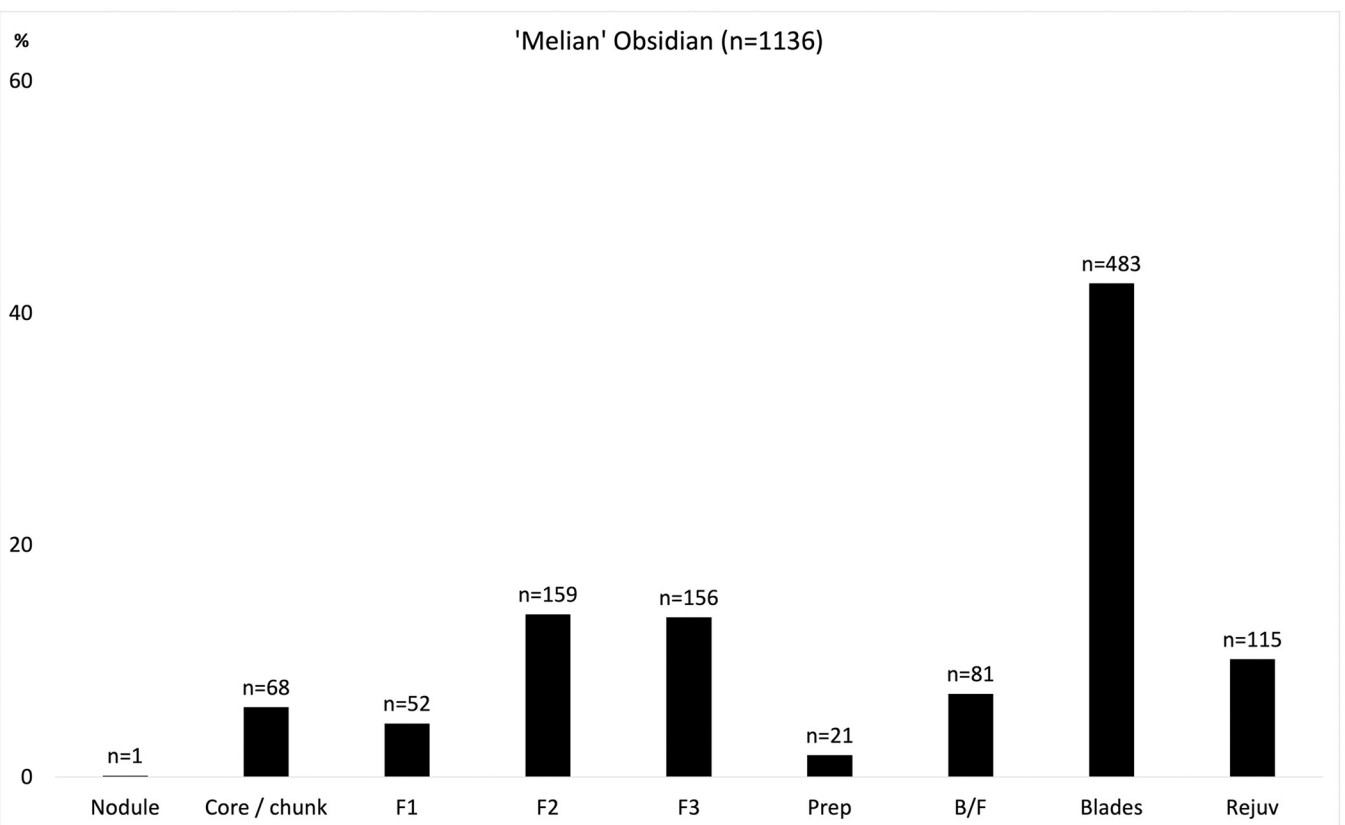

**Fig 6. Techno-typological classes represented in the *Quartier Nu* 'Melian' obsidian assemblage, as defined by visual discrimination.** F1 / F2 / F3 = >80% / 5–80% / <5% cortex on dorsal surface, respectively; Prep = core preparation flake; B/F = blade-like flake; Rejuv = core rejuvenation flake. Original copyright with the authors.

Melian sources: Sta Nychia (Adhamas) and Dhemenegaki [31]. While these early studies were undertaken via Bradford University's School of Applied Physics, most subsequent NAA obsidian sourcing was performed at the National Centre of Scientific Research "Demokritos", Athens [36, 120–122]. The Demokritos research reactor shut in 2004, with NAA analyses of Aegean obsidian subsequently undertaken in Pavia, Italy [123, 124].

The lab analysis of the *Quartier Nu* material began with removing a small flake from each artifact, the piece then etched in hydrofluoric acid (1N) for ten minutes to remove the outer surface. The flake was then ground and a fraction of ~100 mg was weighed for analysis. This grain size is small enough to enable samples to take the shape of the vials and avoid geometry problems during measurements. It is also large enough to avoid handling difficulties of powdered samples, as for example the powder's attachment to the vial wall. Each sample was then heat-sealed in polyethylene vials and irradiated at the Demokritos swimming pool reactor with a thermal neutron flux of $2.7 \cdot 10^{13}$ n · cm$^{-2}$ · s$^{-1}$.

Along with the artifact samples two standards were irradiated (both at 30 minutes): a primary NIST standard, Obsidian Rock, and a check USGS standard, AGV-1. After irradiation, the γ-spectra of the activated samples were measured twice. The first measurement was taken eight days after irradiation, for the determination of Sm, Lu, U, Yb, Sb, Na plus La by integrating the peaks of the respective isotopes, in relation to the standards. Three weeks later the radioactivity had decayed considerably, at which point the samples were measured again for the determination of Ce, Th, Cr, Hf, Ba, Cs, Sc, Rb, Fe, Zn, Co, and Eu. All elemental

**Table 2. Contextual, techno-typological, source, and blind test data for the analysed *Quartier Nu* artifacts.**

| NAA ID | Dig ID | Locus | Building /Area | Deposit Date | Debitage Category | Source | Blind Test |
|--------|--------|-------|----------------|--------------|-------------------|--------|------------|
| MAN01 | CS 354 | 0535 GB 140+b.N. | XIV | LM III | Cortical flake (F2) | Dhemenegaki | Wrong (SN) |
| MAN02 | CS 320 | 0523/9 GC 133W | IV,1 | LM IIIB | Blade—prismatic | Dhemenegaki | Correct |
| MAN03 | CS 392 | 2008 GB 134 | II,5 | LM III | Blade—prismatic (retouched) | Dhemenegaki | Wrong (ANT?) |
| MAN04 | no number | 4231/1 GD 133 | zone VI (sud) | LM III/mod | Cortical flake (F1) | Giali A | Correct |
| MAN05 | no number | 1016/4 GB 135 | XI,6 | LM IIIB/mod | Flake | Giali A | Correct |
| MAN06 | CS 417 | 0510/5, GB 132 | II,9 | LMIIIB | Cortical flake (F2) | Dhemenegaki | Correct |
| MAN07 | CS 344 | 0525/3.1 GB 140 | XIV | LM III | Blade—prismatic | Sta Nychia | Correct |
| MAN08 | CS 345 | 0525/3.1 GB 140 | XIV | LM III | Blade—remnant cortex | Dhemenegaki | Wrong (SN) |
| MAN09 | CS 353 | 0534 GB 140+b.N. | XIV | LM III | Cortical flake (F2) | Sta Nychia | Correct |
| MAN10 | CS 312 | 0516/1 GC 133 W | IV,1 | LMIIIB | Cortical flake (F2) | Sta Nychia | Correct |
| MAN11 | CS 333 | 0529.1 GC 132 | IV,2 | LM IIIB/mod | Blade-like flake (cortical, notched) | Sta Nychia | Correct |
| MAN12 | CS 396 | 1048/1 FZ 134 | II,1 | LM IIIB | Cortical flake (F2) | Sta Nychia | Correct |
| MAN13 | CS 422 | 4300.3 GA 134 | II,1 | LM IIIB/mod | Blade—prismatic | Sta Nychia | Correct |
| MAN14 | CS 350 | 0530/3.4 GB 140+b.N. | XIV | LM III | Blade—remnant cresting | Dhemenegaki | Correct |
| MAN15 | CS 351 | 0530/3.4 GB 140+b.N. | XIV | LM III | Blade—prismatic | Dhemenegaki | Correct |
| MAN16 | CS 324 | 0530 GC 132 | IV,2 | LM IIIB | Blade—prismatic | Sta Nychia | Correct |
| MAN17 | no number | 1038 GC 135 | IX,7 | LM IIIB/mod | Core—blade | Sta Nychia | Correct |
| MAN18 | CS 314 | 0516/2 GC 133(W) | IV,1 | LM IIIB | Flake | Dhemenegaki | Correct |
| MAN19 | CS 459 | 1018/4 GA 133 | I,1 | LM IIIB/mod | Blade—prismatic | Dhemenegaki | Wrong (SN) |
| MAN20 | CS 457 | 1012/3 GA 133 | I,2 | LM IIIB | Rejuvenation flake | Sta Nychia | Correct |
| MAN21 | no number | 1054–4 GB 135 | XI,6 | LMI-III | Rejuvenation (off core back) | Dhemenegaki | Wrong (ANT) |
| MAN22 | CS 413 | 0554/8 GB 133 | II,6 | LM IIIB | Cortical flake (F1) | Sta Nychia | Correct |
| MAN23 | no number | 4077/1-2 GC 133 | zone V | LM III | Blade—prismatic | Dhemenegaki | Wrong (ANT) |
| MAN24 | CS 515 | 2037/1 GE 133 NO | zone fosse I | LM IIIB | Cortical flake (F1) | Dhemenegaki | Wrong (SN) |
| MAN25 | no number | 0089/1.2 B.C./N.-S. | zone XI,6–7 | LM IIIB | Flake | Dhemenegaki | Correct |
| MAN26 | no number | 0033 sondage 92 C | XI,3 | LM IA | Blade—prismatic (*pièce esquillée*) | Dhemenegaki | Correct |
| MAN27 | CS 174 | 1070–1 GA/GB 136 | XI,1–2 | LM IIIB | Blade—prismatic | Sta Nychia | Correct |
| MAN28 | CS 173 | 1068 GA/GB 136-O | zone XI,1–2 | LM IIIB/mod | Blade-like flake | Sta Nychia | Correct |
| MAN29 | CS 22 | 1081 GB 136-centre | XI,1 | LM IIIB/mod | Blade—crested | Sta Nychia | Correct |
| MAN30 | CS 24 | 1080 GB 136-centre | XI,1 | LM IIIB/mod | Rejuvenation flake | Sta Nychia | Correct |
| MAN31 | CS 26 | 1080 GB 136-centre | XI,1 | LM IIIB/mod | Cortical flake (F2) | Sta Nychia | Correct |
| MAN32 | CS 28 | 1080 GB 136-centre | XI,1 | LM IIIB/mod | Rejuvenation flake | Sta Nychia | Correct |
| MAN33 | CS 65 | 0065.1 GE 137 | XIII | LM IIIB/mod | Blade—prismatic | Dhemenegaki | Correct |
| MAN34 | CS 37 | 1090 GB 136-E | XI,1–6 | LM IIIB/mod | Blade—prismatic | Sta Nychia | Correct |
| MAN35 | CS 188 | 1075–1 GA/GB 136-O | XI,1–2 | LM IIIB/mod | Blade—prismatic | Dhemenegaki | Wrong (SN) |
| MAN36 | CS 347 | 0526 GB 140+b.N. | XIV | LM III | Blade—prismatic | Dhemenegaki | Wrong (SN) |

ANT = Anatolian; SN = Sta Nychia; F1 = >80% cortex on dorsal surface; F2 = 5–80% cortex on dorsal surface; mod = modern surface.

concentrations were determined based on the reference material Obsidian Rock, except for La which was determined using AGV. The gamma-ray spectra of standards and samples were analysed with the program GANAAS of the International Atomic Energy Agency.

## Results: Raw material sources and modes of consumption

The elemental concentrations of the 36 artifacts and the NIST standard are presented in Table 3, along with source assignment and an average percent analytical error for each element; the major constituent of the total error was the contribution of counting statistics.

Source assignment was achieved through comparing the artifact chemical signatures with those of pertinent source samples run by the lab under the same analytical conditions [122].

In this study, a relatively straightforward ternary plot of Fe-Ba-Sc clearly discriminates the *Quartier Nu* artifact trace elemental data into three groups (Fig 7) that correlate with geological source data from Melos-Dhemenegaki (n = 17), Melos-Sta Nychia (n = 17) and Giali A (n = 2). We should note here the discriminative power of Sc, which in the case of Sta Nychia–Dhemenegaki separation is essential, although in absolute numbers the difference is small: 1.6–2.1 ppm respectively. This element is determined routinely by NAA with an accuracy of approximately 2%, hence the importance of this technique in an Aegean context. Furthermore, Sc, Fe, and Ba provide satisfactory discriminative power among the Aegean, central European, Italian and some of the Anatolian sources [121, 125–127].

It is necessary to critically reflect on these results in terms of what they tell us about the larger *Quartier Nu* assemblage, as we cannot take them at face value to infer a 48: 48: 2 ratio between the two Melian and Giali A obsidian. While the artifacts selected for analysis embodied the full range of visual types, we included several pieces that were considered non-Melian based on their colour, banding, translucency, etc.; thus, these potential exotica were over-represented in the total sample (Table 2; S1 Table, S1 Appendix). Here we need to discuss the blind test results and their implications for the composition of the overall *Quartier Nu* obsidian assemblage.

The NAA data has shown that our visual source assignation was correct 75% of the time (n = 27/36); while this represents a significant improvement on our *Quartier Mu* blind test results (33% correct, n = 20/60) that were undertaken at the same time [36], it still embodies a notable error. That said, we were again correct in our recognition of Giali A obsidian, corroborating our prior claim that this vesicular raw material is the one Aegean obsidian that can be distinguished visually with confidence. Similarly, we were right with all claims relating to artifacts being made of Sta Nychia obsidian. In turn, as with our *Quartier Mu* study, all the material claimed visually to be 'Anatolian'—because they were slightly more lustrous and/or translucent and/or had red or black bands–turned out to be Dhemenegaki products. It is the Dhemenegaki raw materials which are again shown to be problematic regarding their visual discrimination, with 11/17 pieces misidentified (Table 2).

With the blind test results in mind, the most straightforward implications of our combined visual and elemental characterisation studies are that: (1) over 99% of the obsidian from *Quartier Nu* comes from Melos, (2) less than 1% of the obsidian comes from Giali A. In terms of translating the results from our analysed sample to a total assemblage profile our biggest problem is disentangling the Dhemenegaki and Sta Nychia source materials. When we turn to those artifacts not included in the NAA analysis, there are a further 36 pieces in the database (S1 Table) visually assigned to Dhemenegaki and/or Nenezi Dağ given their translucency and banding that we now feel confident are likely all products of the Dhemenegaki source. To these we must add the 17 artifacts from the elemental analysis, giving us a total of 53 pieces. Allowing for some error in our visual characterisation, we suggest that Sta Nychia raw materials comprise approximately ca. 94.8% of the Final / Postpalatial assemblage, Dhemenegaki ca. 4.6%, while Giali and Göllü Dağ obsidian make up <1% combined. Spatially there does not seem to be any significant pattern concerning how these raw materials were consumed, with products of each source found in association with one another across the site (Fig 4).

## Melian obsidian

**Sta Nychia products.** The 17 artifacts made of Sta Nychia obsidian represent virtually the entire reduction sequence associated with the manufacture of pressure-flaked blades (Fig 8).

**Table 3. Element content and source assignment for the 36 *Quartier Nu* artifacts.**

| Artefact | Source | Sm | Lu | U | Yb | Sb | Na | K | La | Ce | Th | Cr | Hf | Ba | Cs | Sc | Rb | Fe | Zn | Co | Eu |
|---|---|---|---|---|---|---|---|---|---|---|---|---|---|---|---|---|---|---|---|---|---|
| MAN01 | Dhemenegaki | 2.71 | 0.38 | 3.77 | 2.01 | 0.2 | 29917 | 28350 | 22.5 | 38.4 | 13.3 | 2.74 | 4.00 | 664 | 3.76 | 2.22 | 115 | 10575 | 34.1 | 1.12 | 0.53 |
| MAN02 | Dhemenegaki | 2.91 | 0.39 | 3.63 | 2.32 | 0.3 | 30029 | 30126 | 22.0 | 38.8 | 13.1 | 4.10 | 3.99 | 630 | 3.75 | 2.31 | 117 | 10864 | 37.0 | 1.43 | 0.61 |
| MAN03 | Dhemenegaki | 2.80 | 0.39 | 3.36 | 2.25 | 0.2 | 30002 | 25307 | 22.8 | 37.1 | 13.1 | 2.05 | 3.85 | 697 | 3.99 | 2.33 | 113 | 10550 | 28.5 | 1.13 | 0.43 |
| MAN06 | Dhemenegaki | 2.76 | 0.38 | 3.37 | 2.23 | 0.3 | 29154 | 32045 | 21.9 | 37.0 | 12.6 | 2.77 | 4.02 | 647 | 3.35 | 2.24 | 117 | 10313 | 32.4 | 1.21 | 0.56 |
| MAN08 | Dhemenegaki | 2.66 | 0.37 | 3.36 | 2.15 | 0.2 | 29275 | 27395 | 21.9 | 37.7 | 13.1 | 3.02 | 3.69 | 626 | 3.52 | 2.19 | 107 | 10344 | 34.0 | 1.20 | 0.54 |
| MAN14 | Dhemenegaki | 2.78 | 0.40 | 3.74 | 2.24 | 0.3 | 30739 | 26320 | 22.8 | 38.9 | 13.6 | 2.20 | 3.95 | 655 | 3.86 | 2.27 | 113 | 10560 | 35.8 | 1.12 | 0.54 |
| MAN15 | Dhemenegaki | 2.81 | 0.38 | 3.50 | 2.23 | 0.2 | 30617 | 29671 | 22.8 | 39.2 | 13.6 | 3.26 | 4.03 | 686 | 4.04 | 2.34 | 115 | 10869 | 34.0 | 1.21 | 0.53 |
| MAN18 | Dhemenegaki | 2.91 | 0.39 | 3.46 | 2.30 | 0.3 | 29433 | 36628 | 22.9 | 38.1 | 13.4 | 6.52 | 4.06 | 625 | 4.61 | 2.31 | 117 | 11478 | 32.3 | 1.37 | 0.71 |
| MAN19 | Dhemenegaki | 2.89 | 0.39 | 3.45 | 2.34 | 0.2 | 29283 | 32270 | 22.0 | 37.1 | 12.7 | 2.80 | 3.78 | 660 | 3.62 | 2.43 | 115 | 10692 | 29.3 | 1.34 | 0.60 |
| MAN21 | Dhemenegaki | 2.70 | 0.37 | 3.26 | 2.19 | 0.2 | 29583 | 26464 | 22.1 | 36.7 | 12.6 | 2.07 | 3.69 | 658 | 3.64 | 2.24 | 110 | 10330 | 24.3 | 1.09 | 0.53 |
| MAN23 | Dhemenegaki | 2.73 | 0.40 | 3.48 | 2.12 | 0.2 | 29879 | 28586 | 22.3 | 38.1 | 13.1 | 3.66 | 3.81 | 654 | 3.70 | 2.22 | 113 | 10716 | 31.8 | 1.10 | 0.55 |
| MAN24 | Dhemenegaki | 2.82 | 0.40 | 3.62 | 2.24 | 0.3 | 31170 | 28311 | 22.9 | 40.1 | 13.9 | 4.54 | 3.95 | 691 | 3.92 | 2.31 | 118 | 10861 | 34.8 | 1.29 | 0.53 |
| MAN25 | Dhemenegaki | 2.60 | 0.37 | 3.54 | 1.95 | 0.2 | 28867 | 28198 | 21.7 | 37.1 | 13.0 | 3.07 | 3.63 | 618 | 3.50 | 2.13 | 108 | 10230 | 34.6 | 1.12 | 0.50 |
| MAN26 | Dhemenegaki | 2.73 | 0.38 | 3.59 | 2.19 | 0.2 | 29996 | 29108 | 22.4 | 38.2 | 13.4 | 2.39 | 3.87 | 642 | 3.76 | 2.19 | 113 | 10372 | 34.0 | 1.09 | 0.53 |
| MAN34 | Dhemenegaki | 2.81 | 0.40 | 3.93 | 2.25 | 0.3 | 29094 | 28280 | 22.1 | 37.2 | 12.9 | 5.42 | 3.84 | 717 | 3.87 | 2.27 | 114 | 10983 | 62.4 | 1.06 | 0.55 |
| MAN36 | Dhemenegaki | 2.87 | 0.39 | 3.58 | 2.32 | 0.3 | 30099 | 33819 | 22.7 | 39.3 | 13.3 | 2.95 | 3.79 | 680 | 3.82 | 2.31 | 123 | 10611 | 20.3 | 1.38 | 0.61 |
| MAN37 | Dhemenegaki | 2.94 | 0.40 | 3.80 | 2.38 | 0.3 | 30246 | 29513 | 23.0 | 39.1 | 13.4 | 3.39 | 4.03 | 693 | 3.85 | 2.36 | 124 | 10862 | 32.6 | 1.04 | 0.62 |
| MAN07 | Sta Nychia | 2.92 | 0.45 | 4.22 | 2.47 | 0.5 | 30088 | 28483 | 23.5 | 40.2 | 14.7 | 2.45 | 3.82 | 675 | 4.05 | 1.64 | 119 | 8208 | 33.6 | 0.65 | 0.56 |
| MAN09 | Sta Nychia | 2.81 | 0.43 | 3.95 | 2.40 | 0.2 | 28673 | 26101 | 22.6 | 39.0 | 14.0 | 4.03 | 3.86 | 667 | 4.01 | 1.65 | 110 | 9091 | 29.8 | 0.66 | 0.51 |
| MAN10 | Sta Nychia | 2.78 | 0.39 | 3.66 | 2.29 | 0.2 | 29086 | 28595 | 22.5 | 39.6 | 13.6 | 2.20 | 3.68 | 704 | 4.03 | 1.63 | 117 | 7996 | 22.6 | 0.47 | 0.49 |
| MAN11 | Sta Nychia | 2.73 | 0.41 | 3.96 | 2.03 | 0.3 | 28674 | 28338 | 22.3 | 38.3 | 13.9 | 2.70 | 3.66 | 654 | 3.77 | 1.56 | 115 | 8022 | 31.6 | 0.58 | 0.50 |
| MAN12 | Sta Nychia | 2.93 | 0.44 | 3.86 | 2.46 | 0.4 | 29459 | 29957 | 23.5 | 40.8 | 14.3 | 7.53 | 4.10 | 691 | 4.30 | 1.70 | 127 | 8852 | 21.4 | 0.74 | 0.53 |
| MAN13 | Sta Nychia | 2.89 | 0.42 | 3.50 | 2.40 | 0.3 | 28160 | 30914 | 22.4 | 39.5 | 13.7 | 6.75 | 3.97 | 648 | 4.04 | 1.72 | 128 | 7594 | 26.2 | 0.71 | 0.72 |
| MAN16 | Sta Nychia | 2.89 | 0.42 | 3.94 | 2.28 | 0.3 | 28498 | 31369 | 22.9 | 38.6 | 13.3 | 2.68 | 3.84 | 700 | 3.87 | 1.68 | 123 | 8170 | 35.4 | 0.69 | 0.55 |
| MAN17 | Sta Nychia | 2.89 | 0.40 | 3.72 | 2.33 | 0.3 | 28709 | 29122 | 22.5 | 38.7 | 13.5 | 2.21 | 3.60 | 673 | 4.04 | 1.66 | 117 | 8022 | 30.2 | 0.61 | 0.53 |
| MAN20 | Sta Nychia | 2.73 | 0.39 | 3.82 | 2.28 | 0.7 | 28044 | 28424 | 22.1 | 36.3 | 13.6 | 3.07 | 3.64 | 601 | 3.68 | 1.61 | 110 | 7996 | 42.7 | 0.64 | 0.50 |
| MAN22 | Sta Nychia | 2.90 | 0.43 | 4.08 | 2.28 | 0.3 | 30065 | 34091 | 23.6 | 40.3 | 14.6 | 2.30 | 3.91 | 691 | 4.19 | 1.66 | 123 | 8414 | 33.6 | 0.61 | 0.50 |
| MAN27 | Sta Nychia | 2.82 | 0.41 | 3.90 | 2.38 | 0.3 | 28269 | 30519 | 22.4 | 37.7 | 13.6 | 3.03 | 3.63 | 654 | 4.18 | 1.63 | 120 | 8243 | 39.1 | 0.59 | 0.41 |
| MAN28 | Sta Nychia | 2.78 | 0.39 | 3.98 | 2.27 | 0.3 | 27878 | 26603 | 21.8 | 36.5 | 13.4 | 3.82 | 3.60 | 705 | 3.82 | 1.57 | 116 | 7824 | 48.2 | 0.56 | 0.61 |
| MAN29 | Sta Nychia | 2.92 | 0.43 | 3.90 | 2.41 | 0.3 | 29764 | 31721 | 23.7 | 41.4 | 14.2 | 2.84 | 3.83 | 680 | 4.40 | 1.76 | 125 | 8713 | 25.9 | 0.49 | 0.53 |
| MAN30 | Sta Nychia | 2.69 | 0.34 | 3.33 | 2.11 | 0.3 | 26598 | 28083 | 20.7 | 37.7 | 13.2 | 3.89 | 3.55 | 659 | 3.85 | 1.63 | 113 | 7901 | 32.0 | 0.63 | 0.53 |
| MAN31 | Sta Nychia | 2.76 | 0.40 | 3.64 | 2.28 | 0.3 | 27947 | 27084 | 21.7 | 36.2 | 13.6 | 3.44 | 3.54 | 623 | 3.80 | 1.58 | 115 | 8292 | 44.7 | 0.58 | 0.30 |
| MAN32 | Sta Nychia | 2.76 | 0.40 | 3.84 | 2.20 | 0.3 | 27900 | 27162 | 21.7 | 35.8 | 13.4 | 2.68 | 3.48 | 661 | 3.64 | 1.59 | 116 | 7663 | 74.6 | 0.81 | 0.43 |
| MAN33 | Sta Nychia | 2.93 | 0.40 | 3.85 | 2.29 | 0.3 | 29208 | 30840 | 22.8 | 38.7 | 13.5 | 2.70 | 3.58 | 682 | 3.96 | 1.66 | 120 | 8107 | 30.2 | 0.67 | 0.53 |
| MAN35 | Sta Nychia | 2.94 | 0.42 | 4.15 | 2.38 | 0.3 | 28395 | 35132 | 23.4 | 43.6 | 14.3 | 3.92 | 3.83 | 678 | 4.16 | 1.71 | 132 | 8641 | 35.5 | 0.66 | 0.32 |
| MAN04 | Giali A | 3.04 | 0.38 | 4.87 | 2.09 | 0.6 | 27870 | 37195 | 33.8 | 49.8 | 17.6 | 3.36 | 3.40 | 1150 | 5.25 | 1.71 | 145 | 7527 | 32.1 | 0.74 | 0.39 |
| MAN05 | Giali A | 3.03 | 0.39 | 5.09 | 1.96 | 0.6 | 28555 | 38007 | 35.1 | 54.6 | 18.4 | 2.98 | 3.62 | 1123 | 5.69 | 1.75 | 145 | 7210 | 30.2 | 0.47 | 0.33 |

Values in ppm: parts per million.

These include part- and non-cortical flakes from the initial reduction of a raw nodule and shaping the core (Fig 9, 1–8), a crested blade (Fig 9, 9), an exhausted core (Fig 9, 10), a series of fine prismatic end-products (Fig 8, 11–15) plus two rejuvenation pieces flaked from the back of a nucleus (Fig 9, 16–17). These products, their scale, their specific techniques of initiation and rejuvenation are typical of knapping traditions at Malia and in Bronze Age Crete more generally [115, 128–132].

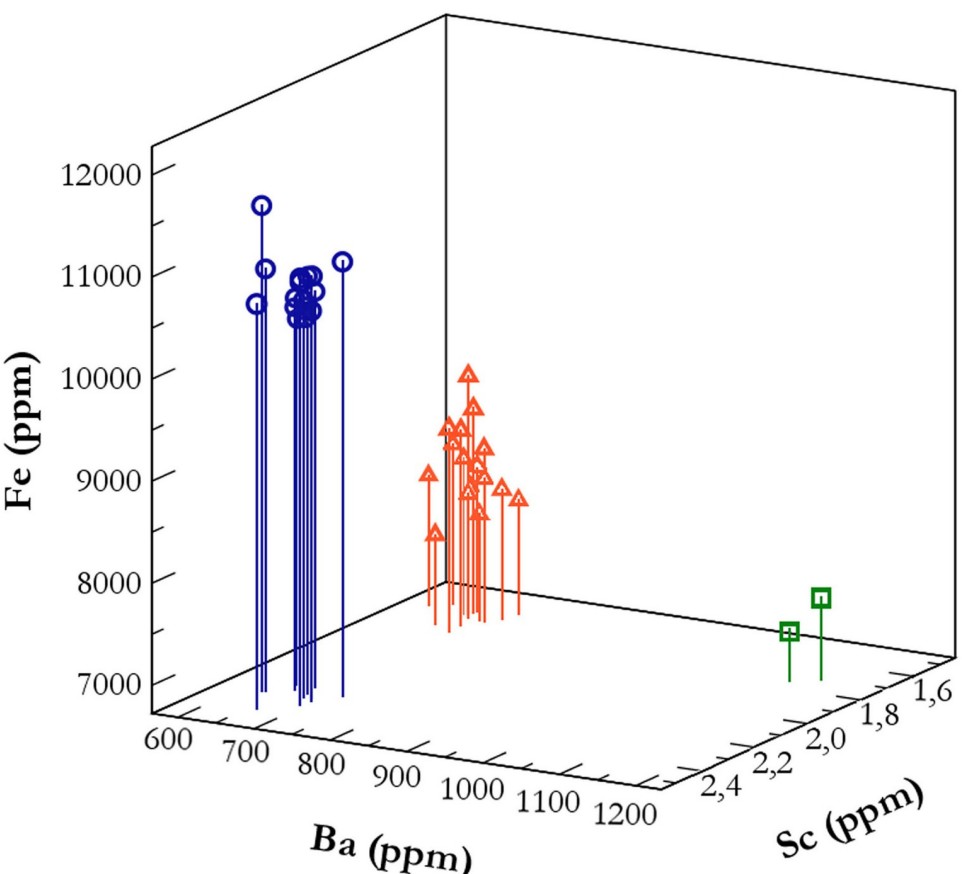

**Fig 7. Ternary Fe-Cs-Sc diagram for the 36 *Quartier Nu* artifacts analyzed using NAA.** Le Bourdonnec, F.-X; original copyright with the authors.

**Dhemenegaki products.** The 17 artifacts characterized as Dhemenegaki obsidian also represent the entire pressure-flaked blade manufacturing sequence (Fig 8), from cortical and non-cortical preparatory flakes (Fig 9, 18–23), via secondary series remnant crested blades (Fig 9, 24) to prismatic end-products (Fig 9, 25–33), plus a rejuvenation flake from the back of a nucleus (Fig 9, 34). This is a distinct form of consumption to that witnessed at Protopalatial (MMII) *Quartier Mu* where Dhemenegaki obsidian seemed to have been procured in the form of part-reduced cores and end-products [36].

Characterisation studies have thus shown that Sta Nychia and Dhemenegaki obsidian was exploited by members of the Malia community for over 1000 years, from EM II–LM IIIB [35, 133]. These sources are now known to have been exploited by overseas populations from at least the Upper Palaeolithic, or the Mesolithic in a Cretan context [28], with the *Quartier Nu* material representing the chronologically latest evidence for the raw materials' exploitation, into the 13[th] century cal. BC.

Malia's preferential access to Melian obsidian–either via direct access voyaging by community members, or intermediary exchange networks–can be traced back until at least the EM IIA period of the mid-3[rd] millennium BC [132]. Prepalatial obsidian workshops are claimed to exist from EM IIB [43, 133, 134], the products of which then being consumed in various local domestic and craft activities, and/or redistributed to members of other settlements / political dependents [135]. This privileged access to obsidian and the skilled pressure-blade makers

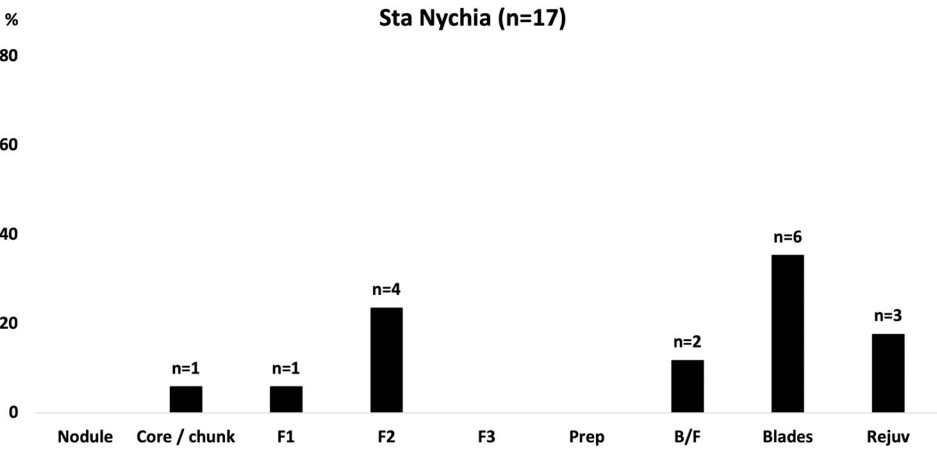

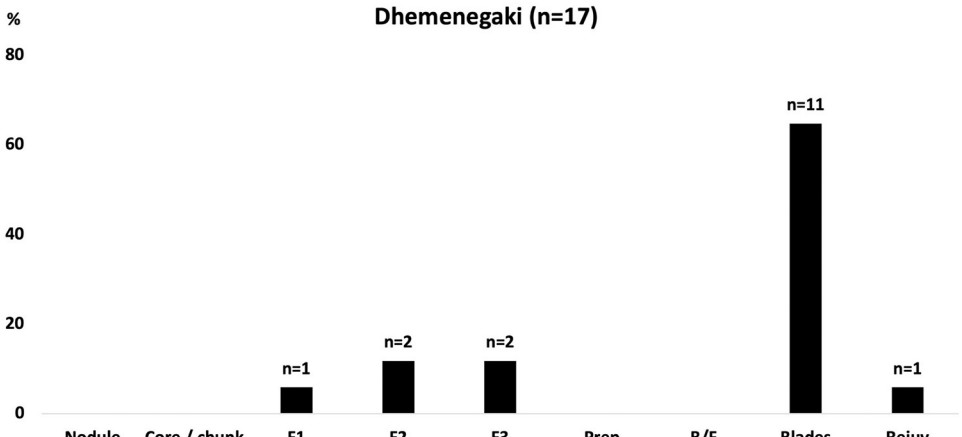

**Fig 8. Techno-typological classes represented in the *Quartier Nu* obsidian assemblage elementally characterized in this study.** Original copyright with the authors.

part-constituted the political capital that enabled members of the community to become local elites [115, 116] whose influence helped transform the site into an 'anomalously large site' of EBA Crete [136].

Significant quantities of Melian obsidian continued to be imported and worked at Malia during the period of the first palaces (MM IB-MM IIB [133, 137]), used for both domestic purposes, and craft activities such as seal-stone production [115]. An ongoing study of the *Batîment Pi* assemblage (MM III/LM IA) attests to the continued consumption of Melian obsidian during the Neopalatial period, though we have little idea as to what is happening in the initial phase of the Postpalatial (LM II-IIIA1), i.e., immediately after the destruction of Malia's palatial complex and major buildings in late LM IA / early LM IB [43].

When we shift to considering *Quartier Nu* in its larger Final Palatial and Postpalatial Cretan context, we are faced with two issues. Firstly, there are few published assemblages of broadly contemporary date, while secondly, obsidian recovered from LM II-IIIB contexts is not always viewed as evidence for continued obsidian use in these periods, with both Blitzer [138] and Evely [139] suggesting that their datasets from LM III Kommos and LM II-III Knossos were largely if not entirely residual. The two datasets we can refer to with some confidence come

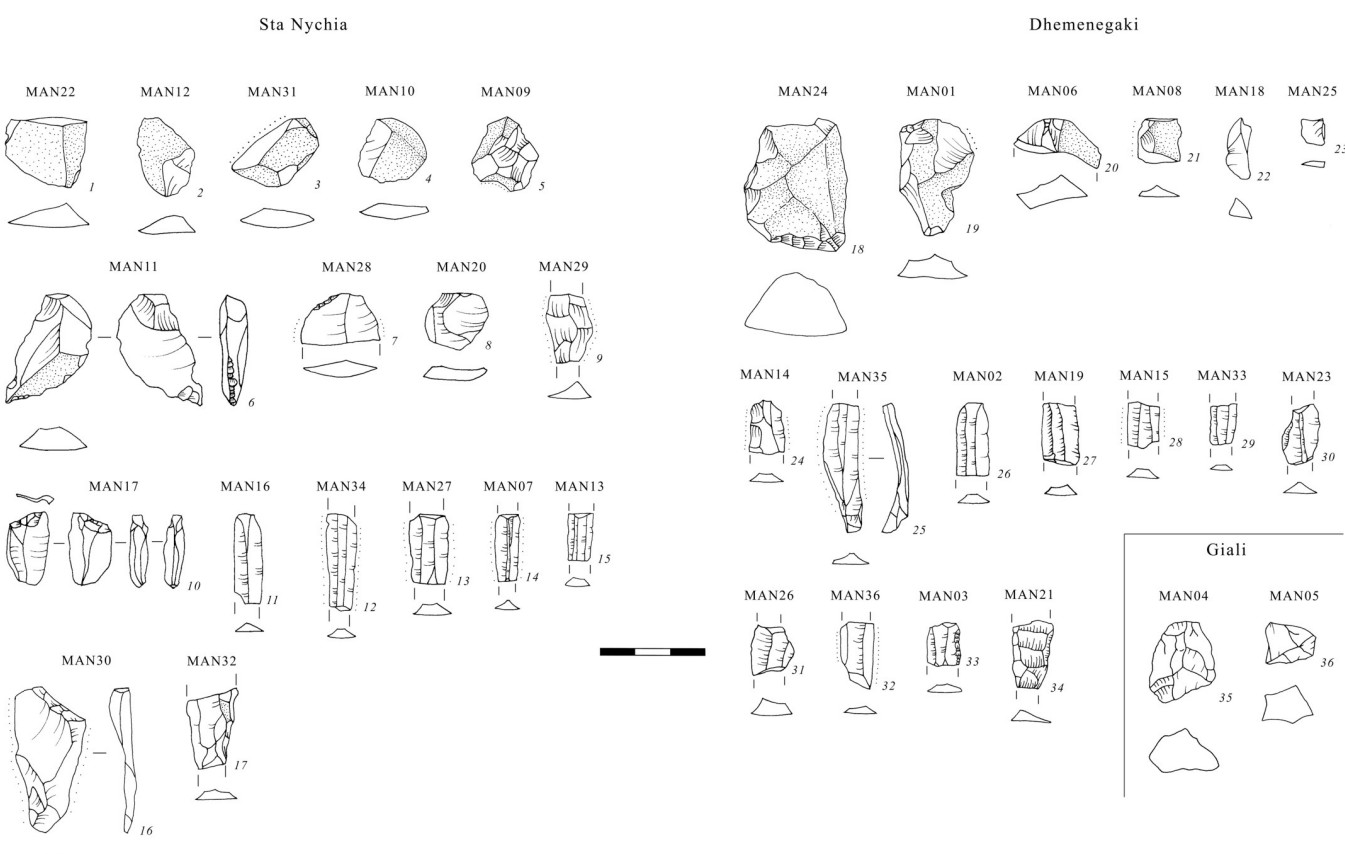

**Fig 9. Artifacts of obsidian sourced to Sta Nychia, Dhemenegaki (Melos), and Giali A.** All cross-sections drawn at midpoint; Labriola, L., Milić, M. Original copyright with the authors.

from Mochlos and Chania. The former, derived from excavations of a LM IIIA1-IIIA2 settlement, with 284 pieces of obsidian [131], 18 of which were elementally characterised in our joint Malia-Mochlos project (all shown to be Melian). The Chania material comprises 684 and 511 artifacts of Melian obsidian from deposits of LM IIIA:2-IIIB:1 and IIIB:2 date respectively [140, 141]. In terms of typo-technological characteristics and the nature of the reduction

**Table 4. Average width and thickness of prismatic blades from Cretan and mainland Middle–Late Bronze Age contexts.** Data for Malia ([115] and this paper), Mochlos [128, 131], Midea and Tsoungiza [143].

| Site | Context | Date | No. | Width (cm) | Thickness (cm) |
|:---:|:---:|:---:|:---:|:---:|:---:|
| Malia | *Quartier Mu* | MM IB-II | 418 (419)* | 0.89 | 0.25 |
| Malia | *Batîment Pi* | MM III-LM IA | 439 (440)* | 0.79 | 0.21 |
| Malia | *Quartier Nu* | LM II-B | 322 | 0.84 | 0.23 |
| Mochlos | Artisans' Quarters | late LM IB | 23 | 0.88 | 0.25 |
| Mochlos | LM III settlement | LM IIIA1-IIIA2 | 88 | 0.83 | 0.24 |
| Midea | Lower Terraces | LH I-II | 8 | 0.85 | 0.26 |
| Midea | Lower Terraces | LH IIIB | 64 | 0.78 | 0.23 |
| Midea | Lower Terraces | LH IIIC | 86 | 0.75 | 0.22 |
| Tsoungiza | Settlement | LH | 23 | 0.93 | 0.29 |

*—sample, not entire assemblage.

sequence, the *Quartier Nu*, Mochlos and Chania assemblages are directly comparable, the communities importing at least some of their obsidian as raw nodules (potentially via their contemporaries at Phylakopi on Melos [142]) that they then shaped, reduced, and rejuvenated in the same manner. The size of the Malia and Mochlos prismatic blades is also much the same (Table 4), while the organisation of production at both these sites was widespread, with several knappers distributed amongst the communities.

**Giali A obsidian.** That two artifacts were shown to be made of obsidian from the Giali A source (Fig 1), came as no surprise, as this is highly distinctive material, its lustrous black hue and white spherulites being characteristics long associated with this Dodecanesian source [34]. MAN04 comprises a thick cortical flake measuring 2.67 × 1.81 cm from zone VI (south), while MAN05, a non-cortical chunk of only 1.64 × 1.23 cm, came from room XI,6 (Fig 9, 35–36). A further three pieces of Giali A obsidian were visually identified from Postpalatial *Quartier Nu*, namely a small chunk from room X,9, a flake from room X,12, and a chip from room XV,2.

While Giali A obsidian was exploited by off-island hunter-gatherer groups from at least the 9th millennium cal. BC onwards [34], its poor fracture habit restricted its use to local communities, with the better-quality Melian obsidian the preferred raw materials for prehistoric tool makers throughout the southern Aegean [113]. It was only during the M/LBA of the 2nd millennium BC that Giali A obsidian came into its own, with skilled Cretan lapidaries crafting it into various finely ground vessels, whose contexts of production and use indicate their elite and / or religious associations [34]. Malia is one of the sites where this raw material has been documented, including three pieces from Protopalatial *Quartier Mu*, one of which came from a seal-stone workshop [36, 115]. Another fragment of Giali A obsidian was characterised in a subsequent project, part of a large obsidian assemblage from MM II House *Delta alpha* at the palace's northwest edge, a dataset that also included a second piece from Giali A, namely a 18 × 11.7 cm cut block thought to be a rough-out for vase manufacture [35, 133, 144]. Recent excavations also produced another small flake from a Protopalatial deposit under *Bâtiment Pi* (T. Carter pers. obs.). Unfortunately, it was impossible to tell whether the flakes of Giali A obsidian from *Quartier Nu* came from sealstone manufacture [145], stone vase production [146], or any other object type.

While most Giali A obsidian from Crete comes from Proto- and Neopalatial contexts [34], a handful of material has been found in Final Palatial deposits at Knossos. A broken fluted and spouted bowl comes from a LM IB-II context from the Royal Road [146], while some small undiagnostic pieces were retrieved from LM II-III deposits in the Unexplored Mansion [139]. On the face of it the *Quartier Nu* material might thus not only represent the latest evidence known for the use of this raw material on Crete (LM IIIB), but also its latest consumption anywhere in the Aegean. That said, given the fragmentary state of all this LM II-III material, and the fact that the Knossian and Malia *Quartier Nu* deposits all overlay Proto- and/or Neopalatial strata, it might be safer to interpret this obsidian as residual, rather than constituting evidence for continuity in socio-economic / crafting practices from the Neopalatial ('Minoan') to Final Palatial ('Mycenaean') worlds. Indeed, it appears that the circulation of Giali A obsidian largely ceased at the end of the Neopalatial period, as new (mainland) tastes and value regimes come to dominate [147], and Aegean trade routes were reconfigured [148].

**Göllü Dağ obsidian.** In 2013, an extra 200 artifacts were recorded that were not included in the original 1999 study. Amongst these pieces were two made of a blue-black translucent obsidian that is associated with the Kaletepe outcrops of the central Anatolian source Göllü Dağ [149] (Fig 1), a distinctive raw material that is visually easy to discriminate [36, 118]. Göllü Dağ was central Anatolia's main obsidian source, exploited at distance from the Upper Palaeolithic to Late Bronze Age [9, 150, 151], with tiny quantities of this material–primarily in the form of ready-made pressure-flaked blades–documented on Crete from the Prepalatial

period. This includes four pieces from EM II Malia [35, 133], five from MM II *Quartier Mu* [36], plus a handful from MM III-LM IA deposits at the nearby *Batîment Pi* (T. Carter, pers. obs.).

The two pieces from *Quartier Nu* comprise a broken non-cortical blade-like flake from a mixed LM I-III (Neo- / Final Palatial) deposit in room X,25 to the east of the site, plus a medial segment of a pressure blade from a context with LM IIIB pottery in room XI,5 to the north of the complex (Fig 4). The latter piece is of particular interest as it was retouched into a trapezoidal form, i.e., a transverse arrowhead, a rare and distinctive weapon in the Aegean [130]. As with the Giali A obsidian from *Quartier Nu* we believe that the central Anatolian pieces comprise residual Neopalatial material (one comes from a mixed deposit), as this obsidian has not been documented previously from any other LBA II-III Aegean contexts, while trapezes of Göllü Dağ obsidian are also known in LM IA deposits in the neighbouring *Batîment Pi* (Fig 3), and Papadiokambos in eastern Crete (Fig 2).

## Traditions of obsidian consumption at Malia: Before and after the LM IB destructions

We turn now to the study's primary research question: Do the raw material and technical choices embodied within the *Quartier Nu* obsidian assemblage reflect a radical break in tradition from how Maliotes were procuring and working obsidian prior to the site's LM IB destruction? The short answer is *no*, they do not. There is clear continuity from the Proto- and Neopalatial practices in terms of (a) raw material choice, and (b) the nature of production and products.

Ideally any discussion concerning potential Mycenaean influence on flaked stone tool traditions at Malia would involve a detailed comparison of Maliote datasets with broadly contemporary mainland assemblages. Alas, there are few detailed examples of the latter material, with the notable exception of reportage upon the lithics from LH III Midea and Tsoungiza in the Argolid [143, 152]. That said, we can point to a major distinction between the tool kits of Post-Palatial Malia with those of their contemporary Mycenaean mainlanders. In providing the evidential bases to support an argument of techno-cultural continuity at Malia from LM I–III we draw primarily upon the Proto- and Neopalatial datasets from *Quartier Mu*, and *Batîment Pi* [36, 115], together with other published assemblages from Malia [133, 137].

### (a) Raw material choice

As noted, Melian obsidian is the dominant raw material of Maliote chipped stone assemblages throughout the Bronze Age, typically comprising >95% of the assemblages (Table 5), the other artifacts being made of obsidian from Giali A and central Anatolia, plus local cherts. More specifically, this and other Maliote sourcing studies suggests that this community had a long-term preference for Sta Nychia obsidian [35, 36]. The relative significance of Melian obsidian at later LM Malia is entirely in keeping with what we see at other well-connected Neopalatial–Postpalatial communities of the Cretan north coast (Fig 2), such as Petras, Mochlos, Poros-Katsambas [153], and Chania (Table 5). In contrast, Melian obsidian never dominates assemblages from the LBA Mycenaean mainland to the same extent, with many of these communities producing a third of their chipped stone tools from chert (Table 5). These data evidence distinct raw material choices being made by Bronze Age Cretan and mainland communities, with the *Quartier Nu* knappers clearly following a local tradition that extended back to the earlier 3rd millennium cal. BC [134]. The different significance accorded chert amongst the mainland populations arguably relates to distinctions in constructions of power, and farming traditions, a hypothesis we develop below.

**Table 5. Relative proportion of raw materials (obsidian and chert) in a selection of Maliote and later Bronze Age Greek mainland chipped stone assemblages.**
*Abords Sud-Ouest du Palais* [132], *Quartier Mu* [115]; *Batiment Pi*, *Quartier Nu* (this paper); Petras [154]; Mochlos [128, 131]; Chania Kastelli [140, 141, 155, 156]; Agios Stephanos [157]; Midea, Tsoungiza [143].

| Assemblage | Date | Total | Melian | Giali A | Anatolian | Chert |
|---|---|---|---|---|---|---|
| *Abords SO du Palais* | EM IIB-III | 166 | 92% | - | - | 8% |
| *Quartier Mu* | MM IB-II | 1943 | 96.5% | 0.2% | 0.3% | 3% |
| *Batiment Pi* | EM II-LM IA | 2225 | 96.2% | 0.1% | 0.3% | 3.4% |
| *Quartier Nu* | LM IIIA2-IIIB | 1153 | 97.8% | 0.3% | 0.3% | 1.6% |
| Petras Houses I.1 and I.2 | MM II–LM IA | 58 | 98.2% | - | - | 1.7% |
| Mochlos Artisans' Quarters | Final LM IB | 63 | 100% | - | - | - |
| Mochlos | LM II-III | 308 | 100% | - | - | - |
| Chania Kastelli | LM II-IIIA:1 | 345 | 100% | - | - | - |
| Chania Kastelli | LM IIIA:2-IIIB:1 | 691 | 99% | - | - | 1% |
| Chania Kastelli | LM IIIB:2 | 511 | 100% | - | - | - |
| Chania Kastelli | LM IIIC | 240 | 99.6% | - | - | 0.4% |
| Agios Stephanos (Laconia) | MH-LH III | 1068 | 91% | - | - | 9% |
| Midea (Argolid) | LH I-II | 54 | 61% | - | - | 39% |
| Midea (Argolid) | LH IIIB | 377 | 66% | - | - | 34% |
| Midea (Argolid) | LH IIIC | 424 | 64% | - | - | 36% |
| Tsoungiza (Argolid) | LH | 108 | 63% | - | - | 37% |

## (b) The nature of production and products

While pressure blade manufacture comprised the typical mode of consuming Melian obsidian amongst southern Aegean Bronze Age communities [113], there were regional, contextual, and chronological distinctions in how this knapping tradition was performed (e.g., modes of core preparation/rejuvenation, and blade initiation), and the size of their end-products [114]. In terms of length, the Maliote cores and blades are typically in the 4–5 cm range, comparable to what we see elsewhere in Crete, particularly during the 2[nd] millennium BC [116, 131, 138, 140, 155].

The cores' relatively short length implies that the pressure-flaked blades were likely removed using a simple hand-held tool, the nucleus being held in the other hand, or a simple clamping device [158]. While the Malia knappers sometimes initiated blade removal by using the natural edge of a rectangular nodule as a ridge for the fracture wave to follow (producing cortical blades), they mainly prepared artificial crests to start the process (Fig 9, 4–5), followed by a secondary series of laminar blanks with remnant cresting scars (Fig 9, 6). These modes of initiation are attested in Maliote assemblages throughout the Pre-, Proto-, Neo- and Final Palatial periods (Table 6), i.e., before and after the period of destruction and socio-cultural change.

We can also consider the ways in which Maliote knappers maximised productivity through rejuvenating the blade core, actions that were usually required when the flaking angle between platform and face became difficult to control and led to mistakes, such as hinged or plunged

**Table 6. Modes of blade initiation and core rejuvenation detailed in Prepalatial to Final Palatial obsidian assemblages at Malia.** CB = crested blade; Bl Rem Cr / Cr = blade with remnant cresting / cortex; Rej Plat / Face / Back = rejuvenation flake from core's platform / face / back.

| Assemblage | Date | CB | Bl Rem Cr | Bl Rem Cor | Rej Plat | Rej Face | Rej Back |
|---|---|---|---|---|---|---|---|
| *Abords SO du Palais* | EM IIA–MM IA | X | X | X | X | X | X |
| *Quartier Mu* | MM IB-II | X | X | X | X | X | X |
| *Batiment Pi* | MM III-LM IA | X | X | X | X | X | X |
| *Quartier Nu* | LM IIIA2-IIIB | X | X | X | X | X | X |

terminations. In these instances, either the core's platform was removed (usually by a flake across the upper face and platform, rather than a true tablet), or the face was taken off (usually struck from the side), or the back of the nucleus was removed (flaked from the platform), and a new flaking surface was opened (Fig 9, 8–9). As before, each of these technical practices is documented in the Malia Pre- to Final Palatial obsidian assemblages (Table 6).

A further argument in support of continuity in Maliote knapping traditions can be made based on the similarity of prismatic blade sizes (width/thickness, few whole examples are found) from Proto- to Final Palatial assemblages (Table 4). While prismatic blades from the later LBA Mycenaean mainland are not dissimilar in size to those from *Quartier Nu* (technical specifics alas being largely absent from reports on these datasets), the blades from LH IIIB Midea are narrower and thinner. We also note the long-term tradition at Malia regarding the organisation of production, with knapping being relatively widespread in all periods.

As noted, we can also refer to clear differences in tool kits between Post-Palatial Malia and the contemporary Mycenaean mainland, specifically regarding the presence and absence of denticulates/sickle elements, and arrowheads. These distinctive implements constitute a key component of mainland assemblages, "by far the most common obsidian artifacts" (Parkinson 1999: 96) from the Palace of Nestor and its surroundings at Pylos (Fig 2), but unknown from the *Quartier Nu* assemblage, and nigh-absent from LM III Crete more generally.

Denticulates, made primarily of chert (occasionally obsidian), on both blades, and flakes, are a recurrent feature of mainland chipped stone assemblages throughout the Bronze Age [152, 159–161]. Given that many of the chert examples have macroscopic gloss detailed along their working edges, a distinctive form of use-wear associated with cutting silica-rich plants [162], such implements are often interpreted as 'sickle elements'. In contrast, denticulates are exceedingly rare in Bronze Age Crete [130], with none documented from the EM II–MM II Maliote assemblages of the *Bâtiment Dessenne* [132], or *Quartier Mu* [115], while LM I *Batîment Pi* (LM I) and *Quartier Nu* (LM IIIB) produced single examples made from obsidian pressure blades. Given that chert is available naturally on Crete [163, 164], the obvious conclusion to draw from the rarity of such implements on the island, is that people were using a different harvesting technology than their mainland contemporaries, with perhaps a greater reliance on bronze sickles to reap their cereals [165–167]. Noteworthy in this regard, is the fact that sickles are depicted in both the Cretan hieroglyphic [168] and Linear A scripts, yet these signs do not appear in the Linear B syllabary of the Mycenaean mainland [169].

The other distinctive chipped stone tool type of the Mycenaean mainland is the arrowhead, with a notable increase in the use of projectiles from the MBA [161]. Obsidian, chert (more commonly), and bronze points are well-attested on the LH III mainland, i.e., broadly contemporary with *Quartier Nu* [152, 160, 170], with some of the finest examples from elite funerary contexts [171]. The character and context of these projectiles attest to archery as a socially valued skill, a reflection of male hunter-warrior elites [172, 173]. On Crete, however, we view the creation and performance of alternative power strategies and forms of masculinity during the Proto- and Neopalatial periods [174–176], with precious few projectiles known to the authors from these periods, and none of the mainland types. It is only with the arrival of Mycenean influenced social practices and/or mainlanders that we find evidence for the appearance of archers, though such personnel seem to be largely restricted to the Knossos area, as evidenced by projectiles—albeit almost exclusively of bronze—from some of the stylistically Mycenaean 'warrior graves' [177, 178], and the reference to massive quantities of arrowheads on the Linear B tablets from the palace (one documents 8,640 arrows [179]). The *Quartier Nu* chipped stone assemblage produced no projectiles.

## Conclusions

The results of the *Quartier Nu* characterisation study allow us to argue for significant cultural continuity at Malia regarding the ways in which members of this community procured and worked their obsidian. Given that population change often leads to shifts in quotidian craft practice [180], we suggest that the Mycenaean character of the site in LM III might be viewed primarily in terms of local elites appropriating new, foreign modes of social distinction, or at most that an indigenous population were now being led by a minority non-local (Knossian / mainland) population. A not dissimilar argument has been forwarded as to the character of the LM III community at Mochlos, where a pottery characterisation study that integrated the analysis of form, decoration, and fabrics, showed that the site's Mycenaean features were context-specific, rather than wholescale. If one focused on the cemetery then the received impression was one of a heavily Mycenaeanized arena, whereas the pottery from the Final Palatial settlement displayed significant technical and stylistic continuity from the preceding 'Minoan' period, albeit with the replacement of the Neopalatial storage-serving-drinking set with one from a Mycenaean-inspired tradition [181]. The Neopalatial–Final Palatial change in ceramic assemblages could thus be viewed as the reworking of a long-standing elite practice of ceremonial drinking by introducing new vessels whose shape and decoration made overt references to Knossian and mainland Mycenaean fashions, thus associating the participants with new power structures. An explanatory model for culture change could thus be based upon shifting modes of social distinction, rather than changes in the population's demographic composition [70], though as with the example of *Quartier Nu*, this need not rule out the presence of a few, politically, and culturally influential non-locals residing there.

The relationship between prehistoric material culture and ethnicity is hugely problematic (if not impossible given the concept's discursive fundamentals [105]), with an object's style and form potentially expressing various aspects of socio-cultural identity dependent upon context (time, place, attendees), including age, gender, status, sodality, etc. [81, 82]. It is thus more productive to focus on reconstructing and mapping cultural traditions as evidenced through 'communities of practice' [182], be they at the site-specific, local, or regional level, where identities were part created and maintained through these agents' participation in exclusive political networks / economic systems / ideological practices, relationships that have material consequences in the form of shared modes of production and consumption [109, 135]. Methodologically this is something archaeologists are eminently capable of documenting through integrated characterisation studies [112, 183], something those working with obsidian sourcing analyses are in an excellent position to develop.

## Supporting information

**S1 Table. The *Quartier Nu* chipped stone assemblage: Context, techno-typological attributes, blade measurements, and general raw material classes by chronological period.**
(XLSX)

**S1 Appendix. *Quartier* Nu database abbreviations used in S1 Table.**
(PDF)

## Acknowledgments

We thank the excavation directors Professors J. Driessen and A. Farnoux for access to the *Quartier Nu* material, while the Greek Ministry of Culture provided study permits; the École Française d'Athènes handled our bureaucratic requests and allowed the reproduction of site plans. We further acknowledge the critical feedback of the two anonymous reviewers, plus

Nicolas Kress, Laura Labriola, Marina Milić, and François-Xavier Le Bourdonnec for the other illustrations. Lucia Alberti, Andy Bevan, Maud Devolder, Don Evely, Ellery Frahm, Charlotte Langohr, Rose Moir, and Angus Smith provided feedback and unpublished data. The paper and conclusions reflect the authors' views alone.

## Author Contributions

**Conceptualization:** Tristan Carter.

**Data curation:** Tristan Carter.

**Formal analysis:** Tristan Carter, Vassilis Kilikoglou.

**Funding acquisition:** Tristan Carter.

**Investigation:** Tristan Carter, Vassilis Kilikoglou.

**Methodology:** Tristan Carter, Vassilis Kilikoglou.

**Project administration:** Tristan Carter.

**Resources:** Tristan Carter, Vassilis Kilikoglou.

**Supervision:** Tristan Carter.

**Validation:** Tristan Carter, Vassilis Kilikoglou.

**Visualization:** Tristan Carter.

**Writing – original draft:** Tristan Carter, Vassilis Kilikoglou.

**Writing – review & editing:** Tristan Carter, Vassilis Kilikoglou.

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
