## [Decision Letter · Decision Letter 0]

20 Apr 2022

PONE-D-22-06920Raw Material Choices and Technical Practices as Indices of Cultural Change: Characterizing Obsidian Consumption At ‘Mycenaean’ Quartier Nu, Malia (Crete)PLOS ONE

Dear Dr. Carter,

Thank you for submitting your manuscript to PLOS ONE. After careful consideration, we feel that it has merit but does not fully meet PLOS ONE’s publication criteria as it currently stands. Therefore, we invite you to submit a revised version of the manuscript that addresses the points raised during the review process.

We look forward to receiving your revised manuscript.

Kind regards,

Peter F. Biehl, PhD

Academic Editor

PLOS ONE

Journal Requirements:

4. We note that Figures 1-4 in your submission contain map/satellite images which may be copyrighted. All PLOS content is published under the Creative Commons Attribution License (CC BY 4.0), which means that the manuscript, images, and Supporting Information files will be freely available online, and any third party is permitted to access, download, copy, distribute, and use these materials in any way, even commercially, with proper attribution. For these reasons, we cannot publish previously copyrighted maps or satellite images created using proprietary data, such as Google software (Google Maps, Street View, and Earth). For more information, see our copyright guidelines: http://journals.plos.org/plosone/s/licenses-and-copyright.

a) You may seek permission from the original copyright holder of Figures 1-4 to publish the content specifically under the CC BY 4.0 license.  

5. Please upload a copy of Figure 9, to which you refer in your text on page xx. If the figure is no longer to be included as part of the submission please remove all reference to it within the text.

6. Please include your tables as part of your main manuscript and remove the individual files. Please note that supplementary tables (should remain/ be uploaded) as separate "supporting information" files.

Additional Editor Comments:

All comments need to be addressed before re-submission.

Reviewers' comments:

Reviewer's Responses to Questions

**Comments to the Author**

1. Is the manuscript technically sound, and do the data support the conclusions?

Reviewer #1: Yes

Reviewer #2: Yes

2. Has the statistical analysis been performed appropriately and rigorously? 

Reviewer #1: I Don't Know

Reviewer #2: Yes

3. Have the authors made all data underlying the findings in their manuscript fully available?

Reviewer #1: No

Reviewer #2: Yes

4. Is the manuscript presented in an intelligible fashion and written in standard English?

Reviewer #1: Yes

Reviewer #2: Yes

5. Review Comments to the Author

Reviewer #1: The theoretical and methodological positioning is clearly presented. It is regrettable that, concerning the notion of the lithic technology ‘chaîne opératoire’, the references do not refer to one of the founding publications such as Tixier et alii 1980, Préhistoire de la pierre taillée 1. Similarly, if the approach implemented does concern technology, sourcing and typology, the use of the term 'functional' is abusive here, since no functional study in the proper sense of the term (use-wear analyses) seems to have been carried out. Note that the illustration in figure 5 has already been published in exactly the same way in Carter 2004a: fig. 21.6.

The historical and archaeological context, local, regional and supra-regional, as well as the problematic and the objectives are precise and complete. Some repetitions from one part to another could be removed, i.e. at the beginning of the section Charting… concerning aims, type of study and questions in debate.

Despite a rather weak corpus, the exploitation of the results seems convincing, but the lack of availability of all the data tables (6 tables) makes it impossible to be sure. In general, the conclusions drawn from the results seem well argued. It would have been interesting to enrich synchronic comparisons of material choices and production techniques with assemblages from 'normal' contemporary sites; this would have supported hypotheses of privileged access to obsidian from Melos, redistribution of obsidian products, and skilled-pressure blade makers as part of political capital involved in access to elite status.

In order to improve the paper some modifications and corrections are required:

- fig. 4: add the north and a scale, indicate the main areas described in the text (court, kitchen, wings)

- fig. 8: complete the lithic drawings with the drawing of the platform and the profile, the preparation of the removal, and the technical conventional symbols (direction of blank knapping, place of the cross-section, etc.). In their current form, the drawings do not support the technological analysis. It could be beneficial to complete them with other pieces of strong technical information, such as cores, whole blades, crested blades, even if they have not been the subject of a NAA.

- Delete references listed in the bibliography but not in the text: Carter et al 2012; D’Annibale 2013; Driessen 1994; Farnoux 1997; Georgiadis 2008; Hood 1990; Karantzali 2016; Kardulias 1992

- Add references cited in the text but not included in the bibliography: Barth 1969; Carter 2007; Jones 2007; Roddick & Stahl 2016; Shennan 1984

- Correct references in the text that are different from the bibliography: Broodbak & Kiriatzi 2007 or 2008?; Carter 2004: a or b ?; Driessen & Farnoux 1994 a or b?; Kiriatzi & Knappett 2016 or Knappett & Kiriatzi 2016?; MacGillivray 2001 or 2000?

Reviewer #2: Overall, this is a fine detailed study. At first, I wondered why it was based only on 36 artifacts, but saw later that those from Quartier Nu were being compared with 60 from Quartier Mu. The incorporation of characterization with sourcing and the overall chaine operatoire are important. And while these were not randomly selected from the total obsidian assemblage, it is clear that Dhemenegaki is a very low percentage.

One item that is inconsistent is the number of pieces of obsidian: on p. 14, it first says "...1153 pieces" but then at bottom of p. 15 it has 125/1276. Also on p. 14, there was a "sample of 168 pieces" - how were they selected in that study?

When was Melos occupied, potentially with local production of cores to Crete and mainland Greece? Or was it entirely raw nodules that were quickly acquired and brought "home" to Crete (as suggested on p. 23).

Very significant is the difference in use-wear between Crete and the mainland in obsidian tool types (denticulates, arrowheads). Since wheat harvesting was important in both places, could this be due to the regular availability of chert on the mainland? With 98% of the lithics at Quartier Nu of obsidian, no surprise that it was used for all lithic tasks.

Minor corrections:

p. 4, line 1: the plural of obsidian is obsidian

p. 4, 3rd to last line: add period to "et al"

p. 5, line 3: change "site" to "sites"

p. 5, line 5: change "contribute" to "contributes"

p. 7, line 3: add space so it reads "LM 1B

p. 7, line 9: add period to "et al"

p. 8, line 8: add "of" so that it reads "Sometime around the end of the..."

p. 8, 1st line of 3rd paragraph: delete "by"

p. 9, line 1: add space so it reads "LM IIIB"

p. 11, line 12: add periods so it reads "Orange et al., 2017"

p. 17, 3rd line last paragraph: remove apostrophe so it reads "1970s"

p. 18, line 5: make lowercase "hydrofluoric"

p. 18, line 6: add space so that it reads "~100 mg"

p. 18, 3rd line from bottom: change to "obsidian" (no s at end)

p. 23, line 12: is it necessary to have "LM" three times? Couldn't it be "LM IIIA:2 - IIIB:1 and IIIB:2"?

p. 24, lines 1 and 2: add space before "cm" so that it reads "1.81 cm" and "1.23 cm"

p. 24, 6th to last line: add space so it reads "11.7 cm"

p. 26, line 13: add "in" and a space so it reads "..also known in LM 1A deposits..."

p. 27, line 5: remove second "LM" so it reads "...from LM 1 - III..."

p. 28, line 10: add space so it reads"...the 4-5 cm range..."

p. 31, line 3: remove comma after "mainlanders"

p 32, end of 1st paragraph: change "here" to "there"

6. PLOS authors have the option to publish the peer review history of their article (what does this mean?). If published, this will include your full peer review and any attached files.

Reviewer #1: No

Reviewer #2: No

---

## [Author Response · Author response to Decision Letter 0]

30 Jun 2022

Responses to editor

• Paper has been reformatted according to PLOS One requirements. Table 3 is too wide however, even when the page is set to landscape, so hopefully your people can configure this appropriately.

• We have also corrected Table 3 by recalculating the missing values, fixing decimals according to precision and removing Nd, Ca and Ta which are not mentioned in the text anyway. We have also removed As from the elements determined (page 20, 2nd line) since it was not included in Table 3.

• Permission has been gained for reproducing the excavation plans, figure captions have been updated to include all pertinent information (those by Lopez, Le Bourdonnec, Milić, Labriola were fully paid for, i.e., the ownership now belongs to Carter). 

• Laboratory protocols: the NAA facility is no longer functioning, so we see little point in uploading the protocols to Protocols.io

• Data availability – those data pertaining to the artifacts’ elemental profile, plus their context, and techno-typological characteristics are made fully available in this paper. Those data pertaining to the techno-typological characteristics, and context for the larger assemblage are not available to share publicly until the full excavation monograph has been published by the École Française d’Athènes.

• Reference list – removed those not quoted in text, and conversely added those mentioned in text, that were missing from the bibliography (as spotted by reviewer #1).

• Reference list – a couple of extra references relating to the chaîne opératoire approach were added to the bibliography (as requested by reviewer #1).

• Reference list – a couple of references were cut, as they were considered superfluous when rereading the text (e.g., Davaras 1971; Rehak and Younger 2001).

Responses to Reviewer #1

• Reference to Tixier et al., 1980 (and two other pertinent references to the concept of the chaîne opératoire) have been added as suggested.

• All bibliographic mistakes kindly listed by Reviewer #1 have been corrected, though four of them were referenced in Table captions: Driessen 1994 (Table 1); D’Annibale 2017; Karantzali 2016; Kardulias 1992 (Table 5).

• Their concern about the misuse of the term ‘functional’ is acknowledged, and any such references removed given the lack of a formal use-wear analysis.

• Fig. 4 – as requested, a scale and north arrow have been added. However, rather than add captions as to the location of kitchens etc. (detailing of multi-functional spaces has been the subject of project research, and is referenced: Driessen & Fiasse, 2011; Driessen et al., 2008), we have noted room numbers when mentioning specific features, such as the central court and freestanding kitchen, so that the reader can locate them.

• That Fig. 5 has previously been published is now acknowledged in the figure caption (“reproduced from Carter, 2004a; Fig. 21.6”). I have the rights to this image and its reproduction.

• Fig. 8 – concerns on lithic illustration conventions: (1) we now detail in the figure caption that all cross-sections were drawn at the artifact’s midpoint; (2) alas it is not possible to include the platform and profile (where missing) as the artifacts were part-destroyed during the analysis; (3) we respectfully note that the direction of blank knapping is indicated with reference to how the artifacts are oriented (this is standard), with the platform uppermost (following the Anglo tradition), the addition of extra symbols denoting flake direction we feel to be superfluous, and is not standard in our region of study (even amongst the classically trained French scholars, see the work of Catherine Perlès for example).

• The concern about the lack of available data – the main techno-typological study of the assemblage is being completed at present for the excavation monograph, in the same way that the sourcing and final studies of the Quartier Mu datasets were published dually. A statement to this effect has been added for clarification.

• By extent, while we understand the logic of the requested extra drawings of obsidian artifacts to detail the entire reduction sequence represented at Quartier Nu (cores etc.), we respectfully suggest that (a) at present the paper/data is neatly iterated in terms of retaining focus, (b) for us to show all the artifacts analyzed is in itself something of a rarity in sourcing studies, let alone a representative sample of the larger assemblage, and (c) such material will be detailed fully in the excavation monograph (again we point to the dual publication of the Quartier Mu datasets). We *do* however, now provide a histogram (Fig 6) that details the techno-typological classes of the larger Final Palatial obsidian assemblage.

• Inconsistencies regarding the size of the assemblage have been corrected; the statement about complete blades has been simplified to remove confusion.

• Repetitions – while structurally some repetition is felt to be necessary, we have cut some text to diminish this issue, in line with the reviewer’s concerns.

• Their statement concerning “it would have been interesting to enrich synchronic comparisons of material choices and production techniques with assemblages from 'normal' contemporary” – we agree, however, such a comparison really takes us away from our main foci, and to be honest there is precious little published in detail from Post Palatial sites away from the north coast for us to undertake such comparisons.

Responses to Reviewer #2

• All grammatical / stylistic errors kindly listed by Reviewer #2 have been corrected.

• The question as to whether chert sickle production use on the mainland related to there being greater amounts of that raw material available, is now addressed by the new text and references in the first half of the following line: “Given that chert is available naturally on Crete (e.g., Blitzer, 2004; Brandl, 2010), the obvious conclusion to draw from the rarity of such implements on Crete, is that people were using a different harvesting technology than their mainland contemporaries, with perhaps a greater reliance on bronze sickles to reap their cereals…”

• The question as to “When was Melos occupied, potentially with local production of cores to Crete and mainland Greece?” and whether we are only dealing with the movement of raw materials is addressed by a rewrite of that sentence to read: “In terms of typo-technological characteristics and the nature of the reduction sequence, the Quartier Nu, Mochlos and Chania assemblages are directly comparable, the communities importing at least some of their obsidian as raw nodules (potentially via their contemporaries at Phylakopi on Melos [Renfrew, 2007]) that they then shaped, reduced, and rejuvenated in the same manner.”

---

## [Decision Letter · Decision Letter 1]

3 Aug 2022

Raw Material Choices and Technical Practices as Indices of Cultural Change: Characterizing Obsidian Consumption at ‘Mycenaean’ Quartier Nu, Malia (Crete)

PONE-D-22-06920R1

Dear Dr. Carter,

We’re pleased to inform you that your manuscript has been judged scientifically suitable for publication and will be formally accepted for publication once it meets all outstanding technical requirements.

Kind regards,

Peter F. Biehl, PhD

Academic Editor

PLOS ONE

Additional Editor Comments (optional):

Reviewers' comments:

Reviewer's Responses to Questions

**Comments to the Author**

1. If the authors have adequately addressed your comments raised in a previous round of review and you feel that this manuscript is now acceptable for publication, you may indicate that here to bypass the “Comments to the Author” section, enter your conflict of interest statement in the “Confidential to Editor” section, and submit your "Accept" recommendation.

Reviewer #1: All comments have been addressed

Reviewer #2: All comments have been addressed

2. Is the manuscript technically sound, and do the data support the conclusions?

Reviewer #1: Yes

Reviewer #2: Yes

3. Has the statistical analysis been performed appropriately and rigorously? 

Reviewer #1: Yes

Reviewer #2: Yes

4. Have the authors made all data underlying the findings in their manuscript fully available?

Reviewer #1: Yes

Reviewer #2: Yes

5. Is the manuscript presented in an intelligible fashion and written in standard English?

Reviewer #1: Yes

Reviewer #2: Yes

6. Review Comments to the Author

Reviewer #1: all remarks and comments have been taken into account. The authors have thus either modified the text according to the recommendations made, or responded appropriately to the remarks.

Reviewer #2: The authors have addressed all of my previous requests. Just one minor issue:

In Table 5, should have consistent number of decimal places (one for all?), at least by column.

7. PLOS authors have the option to publish the peer review history of their article (what does this mean?). If published, this will include your full peer review and any attached files.

Reviewer #1: No

Reviewer #2: No

---

## [Editor Report · Acceptance letter]

12 Aug 2022

PONE-D-22-06920R1 

Raw Material Choices and Technical Practices as Indices of Cultural Change: Characterizing Obsidian Consumption at ‘Mycenaean’ *Quartier Nu*, Malia (Crete) 

Dear Dr. Carter:

I'm pleased to inform you that your manuscript has been deemed suitable for publication in PLOS ONE. Congratulations! Your manuscript is now with our production department. 

Kind regards, 

on behalf of

Dr. Peter F. Biehl 

Academic Editor

PLOS ONE